# Microprism-based two-photon imaging of the mouse inferior colliculus reveals novel organizational principles of the auditory midbrain

Baher A Ibrahim[1,2], Yoshitaka Shinagawa[1,2], Austin Douglas[3], Gang Xiao[1,2,4], Alexander R Asilador[2,3,4], Daniel A Llano[1,2,3,4,5]*

[1]Department of Molecular and Integrative Physiology, University of Illinois at Urbana Champaign, Urbana, United States; [2]Beckman Institute for Advanced Science and Technology, University of Illinois at Urbana Champaign, Urbana, United States; [3]School of Molecular & Cell Biology, University of Illinois at Urbana-Champaign, Urbana, United States; [4]Neuroscience Program, University of Illinois at Urbana-Champaign, Urbana, United States; [5]Carle Illinois College of Medicine, University of Illinois at Urbana-Champaign, Urbana, United States

*For correspondence:
d-llano@illinois.edu

Competing interest: The authors declare that no competing interests exist.

## eLife assessment

This study provides **valuable** new insights into how multisensory information is processed in the lateral cortex of the inferior colliculus, a poorly understood part of the auditory midbrain. By developing new imaging techniques that provide the first optical access to the lateral cortex in a living animal, the authors provide **convincing** in vivo evidence that this region contains separate subregions that can be distinguished by their sensory inputs and neurochemical profiles, as suggested by previous anatomical and in vitro studies. This work provides a foundation for future research exploring how this part of the auditory midbrain contributes to multisensory-based behavior.

**Abstract** To navigate real-world listening conditions, the auditory system relies on the integration of multiple sources of information. However, to avoid inappropriate cross-talk between inputs, highly connected neural systems need to strike a balance between integration and segregation. Here, we develop a novel approach to examine how repeated neurochemical modules in the mouse inferior colliculus lateral cortex (LC) allow controlled integration of its multimodal inputs. The LC had been impossible to study via imaging because it is buried in a sulcus. Therefore, we coupled two-photon microscopy with the use of a microprism to reveal the first-ever sagittal views of the LC to examine neuronal responses with respect to its neurochemical motifs under anesthetized and awake conditions. This approach revealed marked differences in the acoustic response properties of LC and neighboring non-lemniscal portions of the inferior colliculus. In addition, we observed that the module and matrix cellular motifs of the LC displayed distinct somatosensory and auditory responses. Specifically, neurons in modules demonstrated primarily offset responses to acoustic stimuli with enhancement in responses to bimodal stimuli, whereas matrix neurons showed onset response to acoustic stimuli and suppressed responses to bimodal stimulation. Thus, this new approach revealed that the repeated structural motifs of the LC permit functional integration of multimodal inputs while retaining distinct response properties.

## Introduction

Real-world listening occurs in environments cluttered by noise. To navigate these conditions, the auditory system relies on the integration of multiple sources of input. Broad integration of neural inputs, however, comes at a potential cost. Dense convergence can lead to inappropriate cross-talk between input streams, generating a processing 'soup', whereby individual neural pathways cannot be independently modulated. Therefore, highly connected neural systems need to preserve the specificity of inputs in the face of massive integration (*Fair et al., 2007*; *Sporns et al., 2000*; *Tononi et al., 1994*).

Several brain systems have solved the integration/segregation problem by creating repeated structural motifs that allow independent modulation of different pathways. For example, in the basal ganglia, neurochemical periodicity of the striosome and matrix regions permits the retention of functional independence of input streams in the face of massive convergence. As a result, basal ganglia direct and indirect pathways may be independently modulated, while allowing controlled cross-talk between channels (*Alexander et al., 1990*; *Brimblecombe and Cragg, 2017*; *Draganski et al., 2008*; *Gerfen, 1985*).

In the auditory system, the integration/segregation challenge is particularly formidable in the inferior colliculus (IC). The IC serves as a specialized integration hub because of its heavy innervation by higher auditory centers as well as by nonauditory regions. Thus, the IC stands at the crossroads between multiple brain regions and is thus important for linking acoustic stimuli with non-acoustic inputs. The IC comprises a lemniscal division, the central nucleus (ICC), that receives primarily ascending auditory projections (*Oliver, 2005*; *Willard and Ryugo, 1983*; *Willott, 2001*), and two non-lemniscal divisions known as the dorsal (DC) and lateral (LC) cortices that receive massive descending auditory as well as nonauditory input (*Bajo and Moore, 2005*; *Coleman and Clerici, 1987*; *Loftus et al., 2008*; *Saldaña et al., 1996*; *Schreiner and Winer, 2005*; *Winer et al., 1998*) as reviewed in *Lesicko and Geffen, 2022*.

The LC and DC appear to differ in their strategy for integrating these inputs. The DC does not contain known structural heterogeneities but has a tonotopic arrangement of neuronal responses on its surface (*Barnstedt et al., 2015*; *Wong and Borst, 2019*). In contrast, the LC contains repeated neurochemical motifs, here referred to as modules, that express the inhibitory neurotransmitter GABA as well as other metabolic markers (*Chernock et al., 2004*). Areas outside the modules, referred to as matrix, stain strongly for calretinin (*Dillingham et al., 2017*). We and others have determined that nearly all of the long- and short-range connectivity of the LC is governed by the module/matrix organization (*Dillingham et al., 2017*; *Lamb-Echegaray et al., 2019*; *Lesicko et al., 2020*; *Lesicko Alexandria et al., 2016*).

Unfortunately, how the LC integrates these inputs while maintaining distinct paths of information flow is not known. For example, LC neurons respond to multisensory stimuli (*Jain and Shore, 2006*; *Zhou and Shore, 2006*), but information from different modalities enters the LC in segregated channels (*Lesicko Alexandria et al., 2016*). Thus, integration across the module/matrix border must occur, but the mechanisms of this integration remain unknown. Traditional methods of studying the LC, such as in vivo electrophysiology, pose significant challenges in unambiguously assigning individually recorded neurons to residence in matrix or modules. Imaging approaches, such as two-photon (2P) microscopy, have proven to be enormously valuable in understanding the fine spatial structure of neural response properties in the auditory system (*Bandyopadhyay et al., 2010*; *Romero et al., 2020*; *Rothschild et al., 2010*). However, because of its superficial location on the dorsal surface of the mouse midbrain, the DC is the only part of the IC that has been characterized by 2P imaging (*Barnstedt et al., 2015*; *Ito et al., 2014*; *Wong and Borst, 2019*). Optically, it has been impossible to characterize the functions of the LC, which is laterally embedded deep in a brain sulcus.

Here, we describe a novel approach using a 45° microprism inserted into the sulcus that separates the LC from the cerebellum to investigate the fine-scale functional organization of the LC from its lateral surface through 2P imaging. The 2P images obtained from the microprism revealed the first-ever images of the LC in vivo showing its characteristic GABAergic modules. Compared to the DC imaged from the surface, the acoustically and somatosensory-evoked calcium signals obtained by the 2P imaging of the LC through the microprism revealed functional distinctions between the different cellular motifs of the LC. For instance, while modules were more responsive to somatosensory stimulation than matrix, the latter was more acoustically responsive and had a lower detection threshold for complex sounds. Additionally, modules and matrix were found to differently process the spectral and

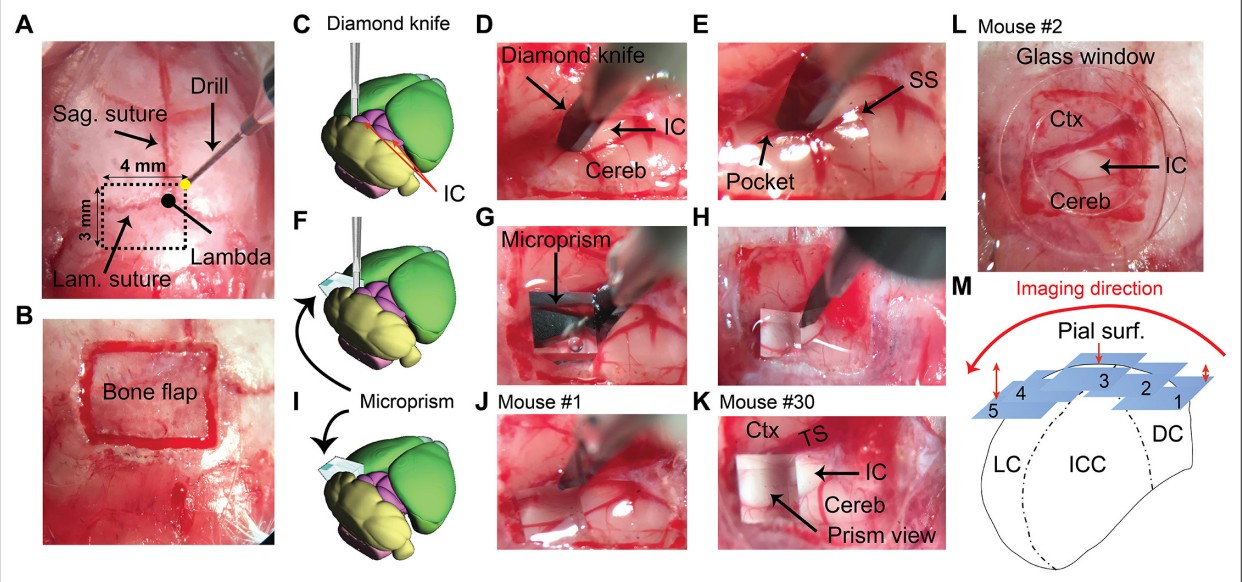

**Figure 1.** A brief illustration of the surgical procedures. (**A**) A surface view of the mouse's skull showing the landmarks of the craniotomy over the IC. (**B**) An image of the bone flap after finishing the drilling. (**C–E**) A cartoon and two images of the brain surface showing the insertion and the pocket made by the diamond knife at the lateral surface of the left IC. (**F–H**) A cartoon and two images of the brain surface show the insertion of the microprism into its pocket to directly face the LC after displacing the blade of the diamond knife. (**I–K**) A cartoon and two images of the brain surface show the final setup of the microprism for two different animals. (**L**) An image of the brain surface shows the placement of a glass window for the surface imaging of the DC. (**M**) A cartoon of the coronal section of the IC demonstrating the different fields of views taken along the mediolateral axis parallel to the curvature of the IC surface at different depths (1–5) from the pial surface. Cereb: cerebellum, Ctx: cerebral cortex, DC: dorsal cortex of the IC, IC: inferior colliculus, ICC: central nucleus of the IC, Lam.: lambdoid, LC: lateral cortex of the IC, Sag.: sagittal, SS: sigmoid sinus, Surf: surface, TS: transverse sinus.

temporal features of the acoustic information. These findings suggest that the non-lemniscal divisions of the IC contain rich but distinct platforms for the processing of different features of auditory and non-auditory information.

## Results

### The insertion of 45° microprism for imaging the LC from the side

GAD67-GFP knock-in mice on a Swiss Webster background were crossed with *Thy1*-jRGECO1a mice to produce a GAD67-GFPx jRGECO1a hybrid mouse, which was used to visualize GABAergic modules, to identify the dense network of GABAergic neurons and terminals in modules of the LC in the green channel (*Tamamaki et al., 2003*) and the calcium indicator, jRGECO1a, in the red channel (*Dana et al., 2018*). A standard craniotomy was performed (*Goldey et al., 2014*) with modifications to suit the specific characteristics of the IC in terms of its location, depth, and high density of the surrounding vasculature (*Xiong et al., 2017*). The craniotomy (~3 × 4 mm²) did not drill directly into the sagittal and lambdoid sutures which allowed minimal disruption of the superficial sinuses below the skull (*Figure 1A*) as indicated by the minimal bleeding near the bone flap (*Figure 1B*). During drilling, pressurized air and a cold stream of saline were used to remove the bone debris and dissipate heat. Because the IC sits deeper in the brain compared to the surrounding structures, the direct insertion of the 45° microprism was found to push the IC ventrally. Therefore, a fine diamond knife was used to separate the lateral surface of the IC from the surrounding tissues without damaging the transverse and sigmoid sinuses (*Figure 1C and D*). This step was followed by a gentle retraction of the knife medially to generate a pocket to secure the microprism (*Figure 1E*). The successful insertion of the microprism was assessed by examining the reflection of the diamond knife from the microprism's top glass surface (*Figure 1F and G*). Holding the microprism down and retracting the knife upward simultaneously resulted in securing the microprism in its pocket (*Figure 1H and I*). Locking the microprism down in its pocket for 10 min with the diamond knife (*Figure 1J*) is critical to ensure that the microprism is stabilized and to prevent bleeding. Filling the gap between the skull surface and

the microprism with saline was found to generate motion artifact due to the heart pulsation, so 1% agarose was used to fill the gap to reduce the motion artifact and add more stability to the microprism (*Figure 1K*). To compare response properties between the LC and DC, direct imaging (i.e. without a microprism) of the dorsal surface of the IC, where the DC is located, was also conducted by scanning the DC surface from the most medial to lateral regions following the convex curvature of the surface (*Figure 1L and M*).

## The 2P characterization of GABAergic and non-GABAergic neuronal distribution on the LC and DC surfaces with histological validation

Following the surgery, the anesthetized animals were taken immediately for imaging. Examination of the distribution of GABAergic modules in the LC was critical in validating the target area for functional calcium imaging. Therefore, the 920 nm laser beam was initially used to scan for GFP, which is expressed in the GABAergic cells of GAD67-GFP knock-in mice. The insertion of the microprism was initially viewed by low-magnification fluorescence imaging of the GFP signals from the whole surface of the IC along with the inserted microprism (*Figure 2A*). This step validated the location of the microprism relative to the IC under the microscope and ensured there was no bleeding that could block the field of view. The 2P images of the GFP signals from the LC imaged via the microprism, or for simplicity LC$_{(microprism)}$, across different animals revealed the LC characteristic GABAergic modules (*Figure 2C and E*, the irregular white lines), which were consistent with previous 3D reconstructions of modules from the coronal sections (*Chernock et al., 2004*; *Lesicko Alexandria et al., 2016*). To further validate the microprism location and the ability to image the LC from the lateral surface, a laser lesion (see Materials and methods) was created via the microprism (*Figure 2G*), which was validated histologically (*Figure 2H*). For comparison, the whole dorsal surface of the IC was scanned from medial to lateral borders (*Figure 2B*, the dotted white lines). Unlike the LC, most of the dorsal surface of the IC showed a non-clustered distribution of the GABAergic cells (*Figure 2D and F*, blue boxes). However, the lateral horizon of the IC dorsal surface showed clusters of GABAergic cells and terminals that were confirmed to be modules, which comprise the prominent feature of the LC (*Figure 2D and F*, red boxes). To further validate the presence of GABAergic modules in the dorsolateral surface of the IC, a laser lesion was created in a GABAergic module located in that area (*Figure 2I and J*, dotted red circle), which was confirmed histologically (*Figure 2K*), demonstrating that it is possible to image a portion of the LC directly from its dorsal surface.

Using the 1040 nm laser, the GAD67-GFPx jRGECO1a hybrid mouse showed a non-clustered cellular expression of jRGECO1a on the surface of the DC (*Figure 3A*) or on the surface of the LC$_{(microprism)}$ (*Figure 3B*), as expected. The IC of anesthetized animals in each experimental setup was then imaged to measure calcium signals over the course of pseudorandom presentations of either 500 ms pure tones of different combinations of frequencies (5–40 kHz, half octave intervals) and sound levels (40–80 dB SPL, 10 dB intervals) or broadband noise without or with amplitude modulation (AM-noise) of different combinations of AM frequencies (2–256 Hz, one-octave intervals) and sound levels (40–80 dB SPL, 10 dB intervals). As shown by two representative cells from the DC imaged from the dorsal surface or LC imaged via microprism, these acoustic stimulation paradigms successfully elicited transient cellular calcium signals that varied depending on the stimulus type (*Figure 3C and D*). The expression distributions of jRGECO1a were histologically examined for the 2P imaged areas of the DC and LC (*Figure 3—figure supplement 1A*). The DC and LC have a similar fraction of cells expressing jRGECO1a (paired sample t-test, $t_{(9)}$ = 0.98, p=0.34, n=10 coronal sections from 2 animals) (*Figure 3—figure supplement 1B and C*). Also, the cells expressing jRGECO1a in all regions of the IC were only non-GABAergic neurons as they did not express GFP which is expressed exclusively in the GABAergic neurons (*Figure 3—figure supplement 1D*).

## Acoustic responsiveness differs in the LC and DC

As indicated by the GFP signals and consistent with *Figure 2*, imaging the LC of the hybrid mice via microprism showed clusters of GABAergic modules, which take different irregular shapes on the surface of the LC (*Figure 4A* – irregular white lines). Based on the best pure tone frequency (BTF) of each responsive cell (*Barnstedt et al., 2015*), an inconsistent and fragmentary tonotopic organization was seen over the LC$_{(microprism)}$ (*Figure 4B*) as indicated by a low $R^2$ of the best linear and quadratic regressions fit made between the BTFs of cells and their locations (*Figure 4—figure supplement 1A*).

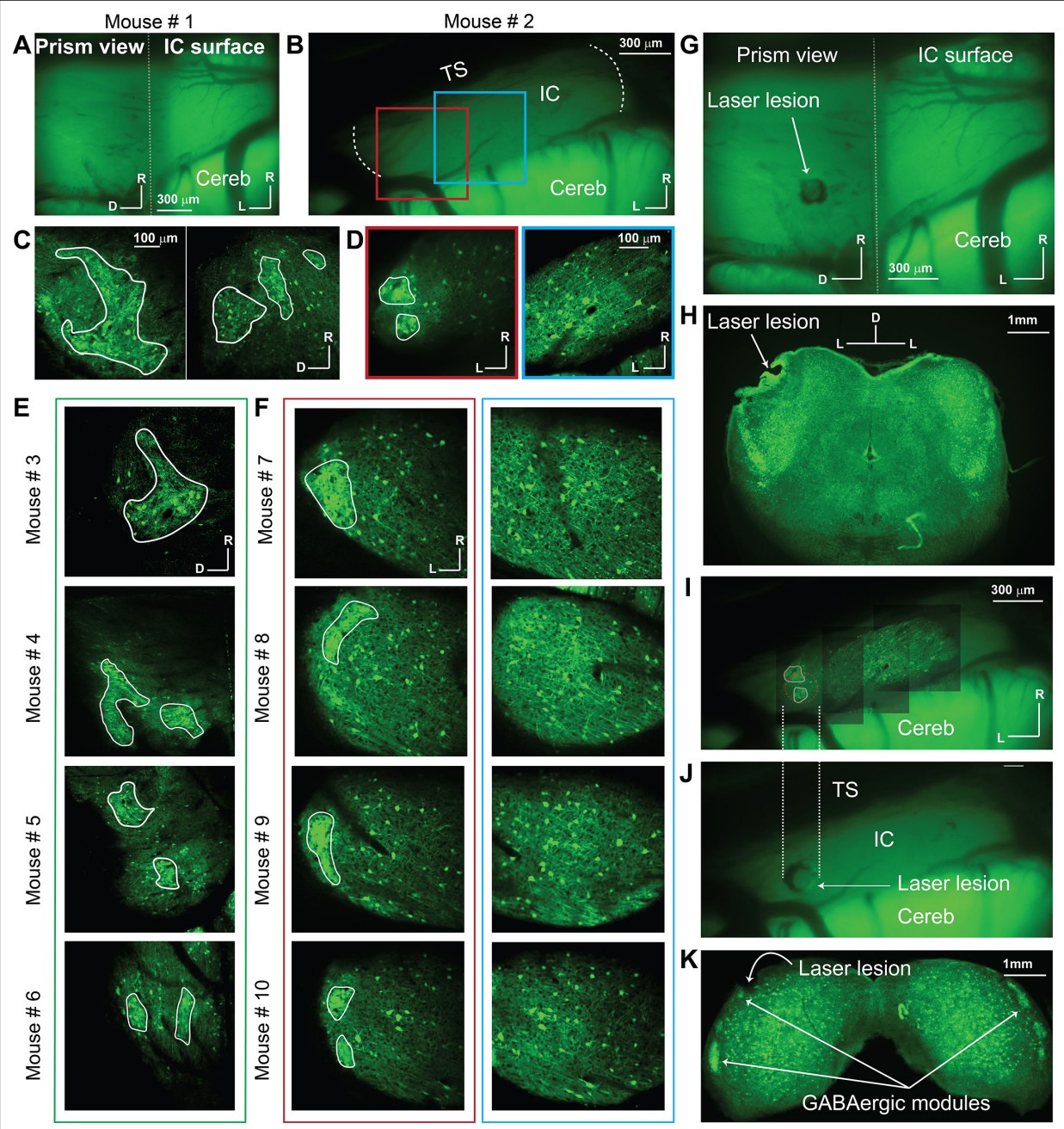

**Figure 2.** The distribution of GABAergic cells on the LC(microprism) and DC imaged from the dorsal surface. All images were taken for the GFP signals exclusively expressed in GABAergic cells. (**A, B**) Low magnification of fluorescence images for the GFP signals obtained from the IC surface with or without the microprism, respectively. The dotted white line in B represented the medial and lateral horizons of the IC. (**C, E**) The two-photon (2P) images of the GABAergic cells on the LC(microprism) show the GABAergic modules across multiple animals. (**D, F**) The 2P images of the GABAergic cells on either the dorsolateral surface of the DC imaged directly from its dorsal surface showing the GABAergic modules across multiple animals (red boxes) or its dorsomedial surface showing a homogenous distribution of GABAergic cells of different sizes with no signs of the GABAergic modules across multiple animals (blue boxes). (**G**) A low-magnification fluorescence image of the IC surface with the microprism showing a laser lesion spot made through the microprism to validate the position of the microprism relative to the IC. (**H**) A histological coronal section at the level of the IC shows that the laser lesion made by the microprism was located at the sagittal surface of the LC(microprism). (**I–J**) Low-magnification fluorescence images of the IC surface from the dorsal surface show a laser lesion spot made on the dorsolateral surface of the LC on the GABAergic module. The dotted white lines represent the alignment of the laser lesion (dotted red circle) to the location of the GABAergic module at the dorsolateral surface. (**K**) A histological coronal section at the level of the IC shows that the laser lesion made on the IC surface targeted the GABAergic module at the dorsolateral portion of the LC. All solid irregular white lines were made at the border of the GABAergic modules. D: dorsal, IC: inferior colliculus, L: lateral, R: rostral, Cereb: cerebellum, TS: transverse sinus.

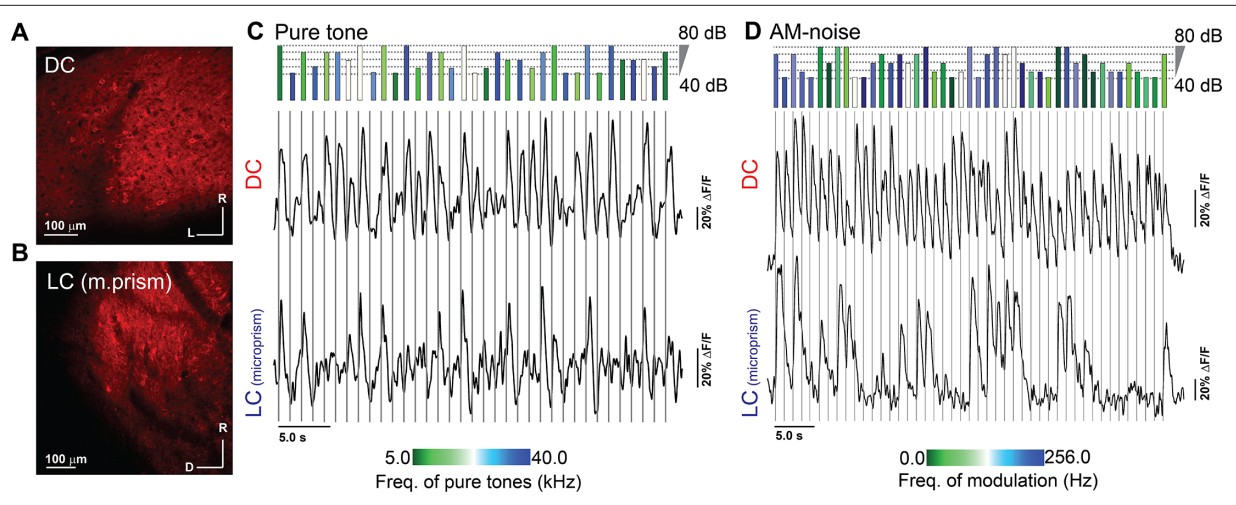

**Figure 3.** Two-photon (2P) expression of jRGECO1a along with evoked signals in the DC imaged from the dorsal surface and LC(microprism). (**A, B**) The 2P images of the jRGECO1a expression on the DC imaged from the dorsal surface and the LC(microprism), respectively. (**C, D**) The time traces of the evoked calcium signals by either different frequencies and amplitudes of pure tones or different overall amplitudes and rates of AM of broadband noise, respectively, which were obtained from two different cells on the DC (top traces) or the LC(microprism) (bottom traces). The length of the colored bars (top) indicates the intensity of the stimulus, while the color indicates the tone or AM frequency depending on the color scale (bottom). AM-noise: amplitude modulated broadband noise, D: dorsal, DC: dorsal cortex, L: lateral, LC: lateral cortex, R: rostral.

The online version of this article includes the following figure supplement(s) for figure 3:

**Figure supplement 1.** The expression of jRGECO1a in the IC across different regions and cell types.

Additionally, the best fit of these regressions was found to be along different axes across the tested animals (***Figure 4—figure supplement 1A***). Despite this inconsistent topographical organization, it was found that the cells in the modules were tuned to relatively higher frequencies than those in the matrix (Mann-Whitney test: z=4.03, p=5.6 × 10⁻⁵, a median of best-tuned frequency = 10 and 7.1 Hz for modules [215 cells from 7 animals] and matrix [230 cells from 7 animals], respectively).

For comparison, the surface of the DC was imaged directly from the dorsal surface using the routine imaging setup. To view the majority of the DC surface, imaging was done by scanning the DC from its medial to the lateral horizon to generate a composite map (see Materials and methods). As indicated by the GFP signals, it was noted that the most lateral region of the surface of the IC showed clusters of the GABAergic cells and terminals (i.e. modules, ***Figure 4C*** – irregular white lines) and are therefore considered part of the LC. Based on the BTF of each responsive cell (***Barnstedt et al., 2015***), the DC showed a tonotopic organization of cells along the rostromedial to caudolateral axis tuned to lower or higher frequencies, respectively (***Figure 4D***), as indicated by a high $R^2$ of the regression fits made between BTFs of cells and their locations along the rostromedial to caudolateral axis (***Figure 4—figure supplement 1B***). It was noted that the $R^2$ of the nonlinear quadratic regression was higher than that of the linear fit. The additional quadratic fit allowed the regression model to include the cells tuned to higher frequencies at the rostromedial part of the DC and result in a better fit, which is consistent with the tonotopic organization that was previously described (***Barnstedt et al., 2015***; ***Wong and Borst, 2019***). It was also noticed that a large area of the DC was tuned to the lowest tested frequency (5 kHz), suggesting a low-frequency bias on the surface of the DC. To quantify tuning differences between DC and LC, the cumulative distribution function based on the cells' BTFs showed a low-frequency bias in the DC, similar to previous work (***Ito et al., 2014***; ***Romand and Ehret, 1990***; ***Shen et al., 2003***; ***Stiebler and Ehret, 1985***). Although around 50% of the cells of LC(microprism) showed a low-frequency bias to 5 and 7.1 kHz, collectively, most of the LC cells were tuned to relatively higher frequencies than those of the DC (***Figure 5A***, Mann-Whitney test: z=–11.34, p<0.001).

Detection of AM is critical in identifying natural sounds as well as in aggregating sound sources during auditory scene analysis, which is important for communication (***Joris et al., 2004***; ***Rees and Langner, 2005***; ***Singh and Theunissen, 2003***). Also, the IC is the first site in the ascending auditory pathway where a substantial number of neurons are rate-tuned to AM (***Frisina, 2001***; ***Joris et al., 2004***; ***Langner, 1992***; ***Nelson and Carney, 2007***; ***Rabang et al., 2012***). Even though calcium

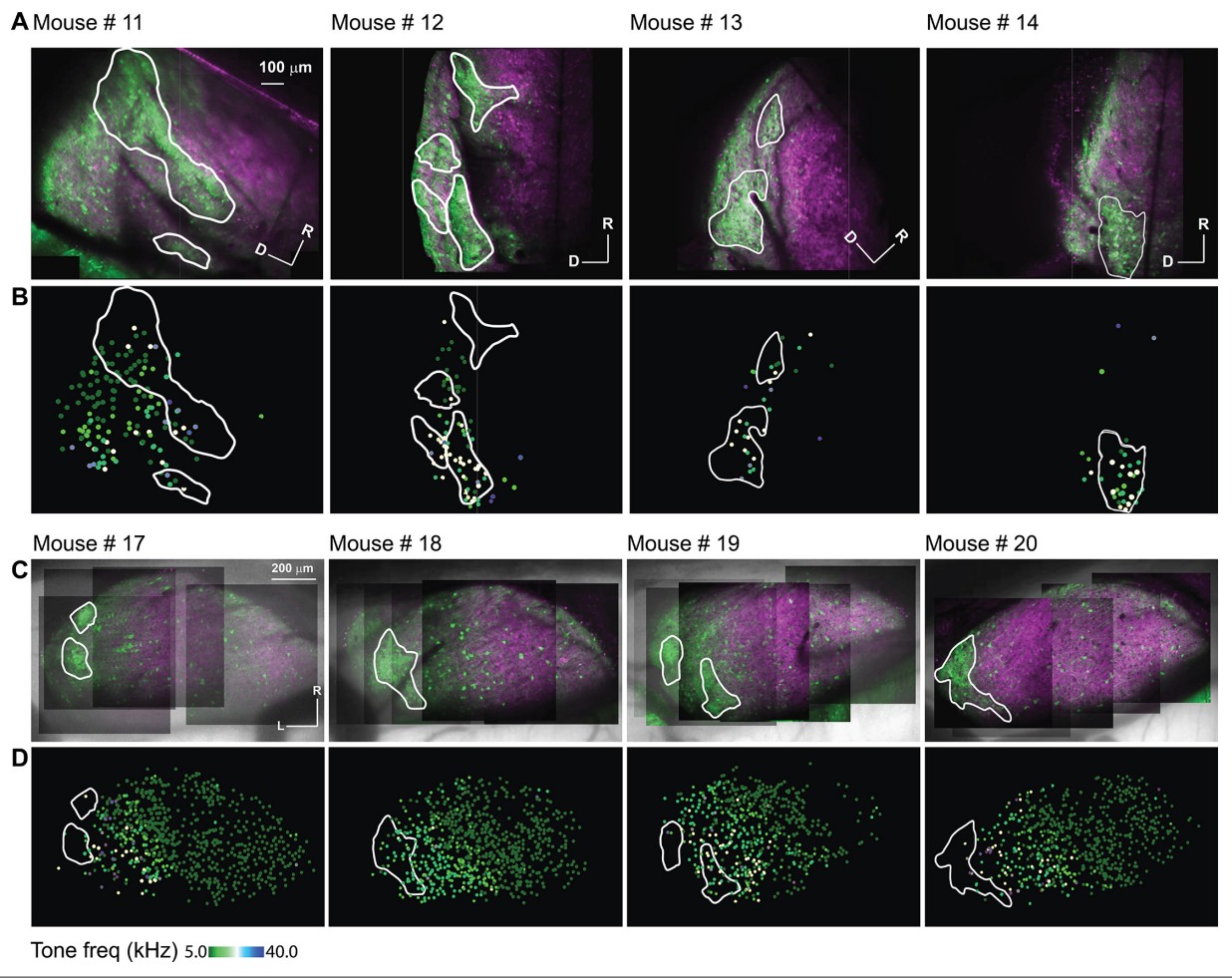

**Figure 4.** The auditory cellular organization of the LC(microprism) vs dorsal cortex (DC) imaged from the dorsal surface. (**A, C**) The two-photon (2P) images of GFP (green) and jRGECO1a expression (red) on either the LC(microprism) or the DC imaged from the dorsal surface, respectively. (**B, D**) The pseudocolor images show the responsive cells to the pure tone of different combinations of frequencies and levels on the surface of the LC(microprism) or the DC imaged from its dorsal surface, respectively. D: dorsal, L: lateral, R: rostral.

The online version of this article includes the following figure supplement(s) for figure 4:

**Figure supplement 1.** The best regression fit between best tone frequency and different anatomical axes.

indicators do not provide a spike-by-spike readout of neural activity, their integrated signal does correlate with the number of spikes produced by a neuron over short time windows (*Vogelstein et al., 2010*; *Yaksi and Friedrich, 2006*). Thus, this integrated signal can be used to measure rate responses to AM. Using this integrated signal, we tested the ability of the cells of the LC(microprism) to respond to broadband noise at different AM rates within each cellular motif. Within the fraction of responsive cells, the LC(microprism) showed cells of different response profiles. While some cells responded only to pure tones or tone-selective (Tone-sel), other cells responded only to AM-noise or noise-selective (Noise-sel). However, some cells responded to both pure tones and AM-noise (Non-sel), when the two stimuli were individually presented. While the LC(microprism) had similar fractions of Noise-sel and Non-sel cells, both fractions were significantly higher than that of the Tone-sel cells (*Figure 5B*, one-way ANOVA, p=1.6 × 10⁻⁴). We also observed that modules and matrix differentially process the spectral information. For instance, the matrix had a significantly greater fraction of Noise-sel cells compared to the modules (*Figure 5C*, Mann-Whitney test, z=−2.32, p=0.02). In contrast, both cellular motifs had similar fractions of Tone-sel and Non-sel cells (*Figure 5C*, Mann-Whitney test: z=0.48, p=0.62 for Tone-sel and z=1.36, p=0.17 for Non-sel). These findings suggest that neurons in the matrix are more responsive to sounds of greater spectral and temporal complexity than those in the modules. Similar

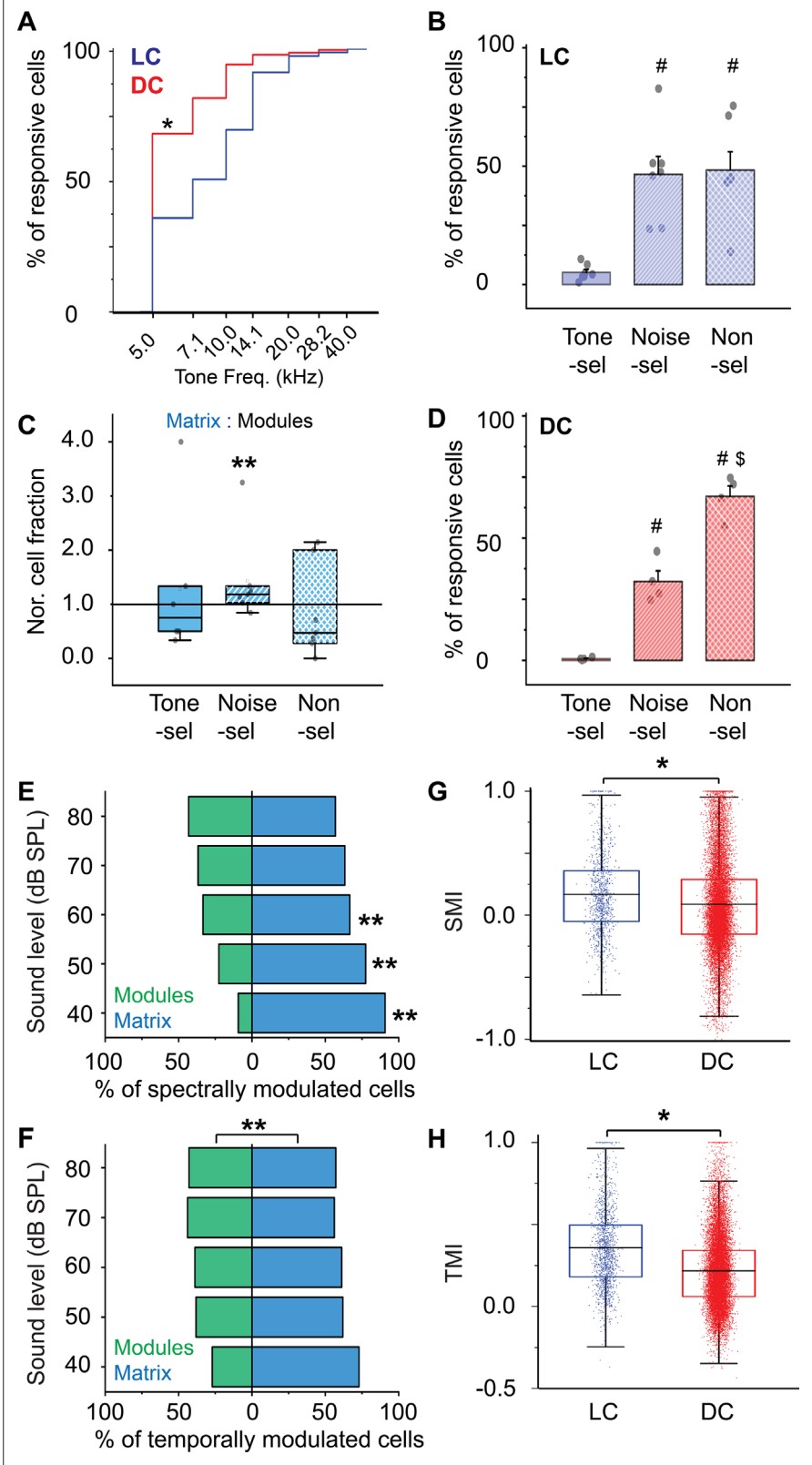

**Figure 5.** The cellular response to pure tones and AM-noise of the LC(microprism) vs DC imaged from the dorsal surface. (**A**) The cumulative distribution function of the best tone frequencies of all cells collected either from the LC imaged via microprism (blue line) or the DC imaged directly from its dorsal surface (red line) (Mann-Whitney test: z=–11.34, p<0.001, a median of best-tuned frequency = 5 and 7.1 Hz for the DC (2484 cells from 4 animals)

*Figure 5 continued on next page*

*Figure 5 continued*

and the LC$_{(microprism)}$ (445 cells from 7 animals), respectively, *p<0.05, DC vs LC$_{(microprism)}$). (**B**) A bar graph showing the fractions of cells responding to only pure tone (Tone-sel), only AM-noise (Noise-sel), or to both nonselectively (Non-sel) on the LC$_{(microprism)}$ (7 animals) (one-way ANOVA, $f_{(2,18)}$ = 14.9, p=1.6 × 10$^{-4}$, Fisher's post hoc test: p=2.0 × 10$^{-4}$ and 1.3×10$^{-4}$ for Noise-sel and Non-sel vs Tone-sel, respectively, p=0.84 for Noise-sel vs Non-sel, % of responsive cells ± SEM = 5 ± 1%, 46 ± 7%, and 48 ± 7% for Tone-sel, Noise-sel, and Non-sel, respectively, #p<0.05 vs Tone-sel). (**C**) Box graphs showing the fractions of Tone-sel, Noise-sel, and Non-sel cells within modules and matrix of the LC$_{(microprism)}$. (**D**) A bar graph showing the fractions of Tone-sel, Noise-sel, or Non-sel cells on the DC imaged directly from the dorsal surface (4 animals) (one-way ANOVA, $f_{(2,9)}$ = 86.9, p=1.2 × 10$^{-6}$, Fisher's post hoc test: p=1.5 × 10$^{-4}$ and 3.4×10$^{-7}$ for Noise-sel and Non-sel vs Tone-sel, respectively, p=7.0 × 10$^{-5}$ for Noise-sel vs Non-sel, % of responsive cells ± SEM = 0.6 ± 0.2%, 32 ± 4%, and 67 ± 4% for Tone-sel, Noise-sel, and Non-sel, respectively, #p<0.05 vs Tone-sel and $p<0.05 vs Noise-sel). (**E, F**) Bar graphs showing the percentage of spectrally (two-way ANOVA: $f_{(1, 4, 20)}$=50.3, 0.0, and 4.0 – p=7.1 × 10$^{-7}$, 1.0, and 0.01 for the region, sound level, and interaction, respectively, Fisher's post hoc, p=5.1 × 10$^{-6}$, 4.9×10$^{-4}$, 0.02, 0.058, and 0.14 for modules vs matrix at 40, 50, 60, 70, and 80 dB, respectively, % of spectral modulated cells ± SEM (modules vs matrix)=9 ± 1% vs 90 ± 1% at 40 dB, 22 ± 4% vs 77 ± 4% at 50 dB, 33±10% vs 66 ± 10% at 60 dB, 36 ±13% vs 63 ± 13% at 70 dB, and 43 ±11% vs 56 ± 11% at 80 dB, n=3 animals) or temporally (two-way ANOVA: $f_{(1, 4, 20)}$=19.3, 0.0, and 1.2 – p=7.1 × 10$^{-7}$, 1.0, and 0.32 for the region, sound level, and interaction, respectively, % of temporally modulated cells ± SEM = 38 ± 3% and 61 ± 3% for modules vs matrix, n=3 animals) modulated cells, respectively, across different sound levels in modules (green bars) vs matrix (blue bars) imaged via microprism (**p<0.05, matrix vs modules). (**G, H**) Box plots showing the mean (black lines) and the distribution (colored dots) of the SMI (two-way ANOVA: $f_{(1, 4, 14583)}$=48.4, 210.4, and 21.3 – p<0.001 for the region, sound level, and interaction, respectively, Fisher's post hoc test, p=6.2 × 10$^{-4}$, 4.2×10$^{-5}$, <0.001, and <0.001 for LC vs DC at 40, 60, 70, and 80 dB respectively, mean of SMI ± SEM: 0.16 ± 0.01 and 0.08 ± 0.003 for LC vs DC, n=1020 cells from 7 animals (LC$_{(microprism)}$), and 13,574 cells from 4 animals (DC)) and TMI (two-way ANOVA: $f_{(1, 4, 16139)}$=431.4, 59.1, and 8.3 – p<0.001 for the region, sound level, and interaction, respectively, Fisher's post hoc test, p=0.002, <0.001, <0.001, <0.001, and <0.001 for LC vs DC at 40, 50, 60, 70, and 80 dB, respectively, mean of TMI ± SEM: 0.35 ± 0.006 and 0.21 ± 0.001 for LC vs DC, n=1441 cells from 7 animals (LC$_{(microprism)}$), and 14,708 cells from 4 animals (DC)), respectively, across different sound levels in the LC$_{(microprism)}$ (blue) vs the DC (red), *p<0.05, LC vs DC. DC: dorsal cortex, LC: lateral cortex. SMI: spectral modulation index, TMI: temporal modulation index.

to the LC, the DC showed significantly higher fractions of Noise-sel and Non-sel cells than Tone-sel cells, but the cellular fraction of Non-sel cells was higher than that of Noise-sel cells (***Figure 5D***, one-way ANOVA, $f_{(2,9)}$ = 86.9, p=1.2 × 10$^{-6}$).

By pooling the cells based on their response to either pure tones or AM-noise, animals showed a lower fraction of responsive neurons to the acoustic stimulation in the LC$_{(microprism)}$ compared to the DC regardless of the sound type (two-way ANOVA, $f_{(1,1,18)}$ = 9.1, 7.8, and 0.1, p=0.007, 0.01, and 0.75 for region [LC vs DC], stimulus [pure tones vs AM-noise], and interaction, respectively, n=7 animals for LC and 4 animals for DC, the % of responsive cells ± SEM = 48±7% for LC and 75 ± 2% for DC). To evaluate if any optical artifacts due to imaging through a prism could be responsible for this finding, the number of responsive cells imaged from the lateralmost dorsal surface of the IC, which we consider part of the LC because of the presence of GABAergic modules (LC imaged from the dorsal surface or LC$_{(dorsal surface)}$), was quantified and compared with that obtained from the LC$_{(microprism)}$. We observed that there were no significant differences between the number of tone or noise-responsive neurons between the LC$_{(microprism)}$ and LC$_{(dorsal surface)}$ (two-way ANOVA, $f_{(1,1, 18)}$=0.07, 14.1, and 2.11, p=0.79, 0.001, and 0.16 for the region, stimulus [pure tones vs AM-noise], and interaction, respectively, n=7 animals for LC$_{(microprism)}$ and 4 animals for LC$_{(dorsal surface)}$, the % of responsive cells ± SEM = 48 ± 7% for LC$_{(microprism)}$ and 45 ± 11% for LC$_{(dorsal surface)}$).

## LC and DC respond differently to spectral and temporal modulation of acoustic inputs

Given that more cells from LC$_{(microprism)}$ and DC had responses to AM-noise compared to pure tones, we further quantified the selectivity to spectral or temporal modulation by computing the spectral modulation index (SMI) or the temporal modulation index (TMI), respectively, as well as the number of modulated cells for all the responsive cells (see Materials and methods) across the tested sound levels.

For the LC$_{(microprism)}$, the matrix has a greater proportion of spectrally modulated cells (cells responding only to pure tones and unmodulated noise) than modules. The higher fractions of modulated cells in

matrix were most prominently at sound levels below 70 dB (*Figure 5E*, two-way ANOVA: p=7.1 × 10⁻⁷, 1.0, and 0.01 for the region, sound level, and interaction, respectively), suggesting that the cells in the matrix have a lower threshold to detect the spectral features of sound. Similarly, the matrix had more cells responsive to temporal modulation (cells responding to AM-noise) as a main effect (*Figure 5F*, two-way ANOVA, p=2.7 × 10⁻⁴, 1.0, and 0.32 for the region, sound level, and interaction, respectively), which was consistent with the above results (*Figure 5C*).

The DC and LC imaged via microprism were found to have different profiles of modulation of responses to spectral and temporal complexity. Although there is a general trend for SMI and TMI to decrease with increasing sound pressure level, corresponding to a decrease in the response to pure-tone and AM-noise, respectively, overall there is a statistically significant higher level of average SMI and TMI in the LC compared to the DC (*Figure 5G and H*, SMI: two-way ANOVA, p<0.001 for all region, sound level, and interaction, TMI: two-way ANOVA, p<0.001 for all regions, sound level, and interactions). These results suggested that neurons in the LC are more sensitive to spectral and temporal modulation than those in the DC.

## LC modules and matrix differentially respond to acoustic and somatosensory stimuli

Somatosensory inputs from many cortical and subcortical structures have been identified in the non-lemniscal division of the IC, particularly in the LC (*Lesicko Alexandria et al., 2016*; *Lohse et al., 2022*; *Zhou and Shore, 2006*). Previous work has shown that while somatosensory projections mostly target the modules, the auditory projections tend to target the matrix (*Lesicko Alexandria et al., 2016*), which indicates that the LC is a multisensory integration site. Therefore, we examined the somatosensory and auditory cellular responses on the surface of the LC by coupling the 2P imaging with the microprism with a whisker or acoustic stimulation (see Materials and methods). Acoustic stimulation was done with 500 ms of unmodulated broadband noise at 80 dB SPL. After the acoustic or somatosensory stimulation, evoked cellular responses were observed from the neurons of the LC(microprism) as well as the DC imaged directly from the dorsal surface. According to these cellular responses, different types of neurons were found in modules and matrix of the LC(microprism) as well as the DC (*Figure 6A–D*). Also, the auditory response evoked by auditory stimulation was only found to be modulated (suppressed vs enhanced) when the auditory stimulation was simultaneously presented with somatosensory stimulation (bimodal stimulation) (*Figure 6E and F*). The cells of every structure were then mapped according to their responses. Auditory and somatosensory responses were evoked in the neurons of both matrix and modules imaged via microprism (*Figure 7A, B, and D*). Within the pool of responsive cells, the cells were categorized into cells that respond only to auditory stimulation (auditory selective cells [Aud-sel]), only to somatosensory stimulation (somatosensory selective cells [Som-sel], or nonselectively to both stimuli [Aud/Som-nonsel]) (*Figure 7F*). Both modules and matrix showed a higher fraction of Aud-sel cells than that of Som-sel and Aud/Som-nonsel cells (*Figure 7H*, chi-square test, $\chi^2$=33.1, p=3.5 × 10⁻⁶). Pooling the cells based on their auditory vs somatosensory responses revealed that more auditory-responsive cells (Aud-sel+Aud/Som-nonsel) were found in the matrix, while more somatosensory-responsive cells (Som-sel+Aud/Som-nonsel) were clustered in the modules (*Figure 7B, D, F, and I*, chi-square test, $\chi^2$=26.0, p=9.3 × 10⁻⁵), which was consistent with the reported difference in inputs to modules compared to matrix (*Lesicko Alexandria et al., 2016*). Additionally, while most of the acoustically responsive cells in the matrix showed onset responses, the cells showing acoustically driven offset responses were clustered in the modules (*Figure 7C and J*, chi-square test, $\chi^2$=120.5, p<0.001). Moreover, a higher fraction of Aud/Som-nonsel cells in the modules had offset, not onset, auditory response than those located in the matrix (chi-square test, $\chi^2$=13.2, p=0.004, % of Aud/Som-nonsel cells = 57%, 15%, 8%, and 20%, for auditory offset cells in modules, auditory onset cells in modules, auditory offset cells in matrix, and auditory onset cells in the matrix, respectively), which suggests a multisensory integrative function by those cells as discussed later. Given the specific spatial location of the somatosensory-responsive cells in the modules which contain clusters of GABAergic cells, we asked if somatosensory stimulation could suppress the auditory response when both stimuli are presented simultaneously (bimodal stimulation). To test this hypothesis, we calculated and visualized the response index (RI), which was calculated as the ratio of the cellular response to bimodal stimulation to the cellular response to acoustic response alone for auditory-responsive cells. Accordingly, some cells showed suppressed (RI<1) or enhanced (RI>1)

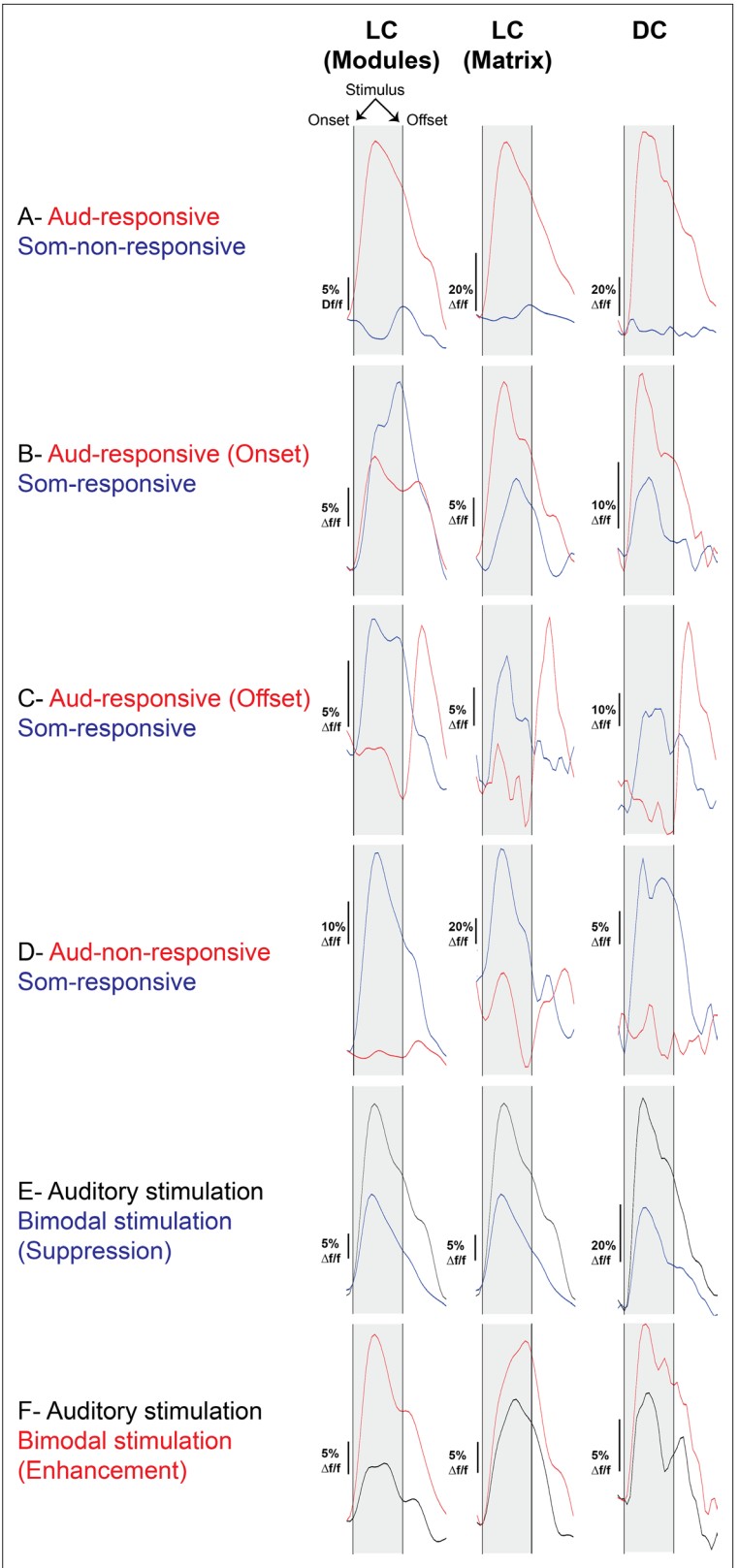

**Figure 6.** The waveforms of the somatosensory vs acoustic responses of the LC(microprism) and the dorsal cortex (DC) imaged via microprism. Examples of different cell types based on their responses to auditory or somatosensory stimulations as well as their response timing to the auditory stimulation as indicated by waveforms of the evoked calcium signals.

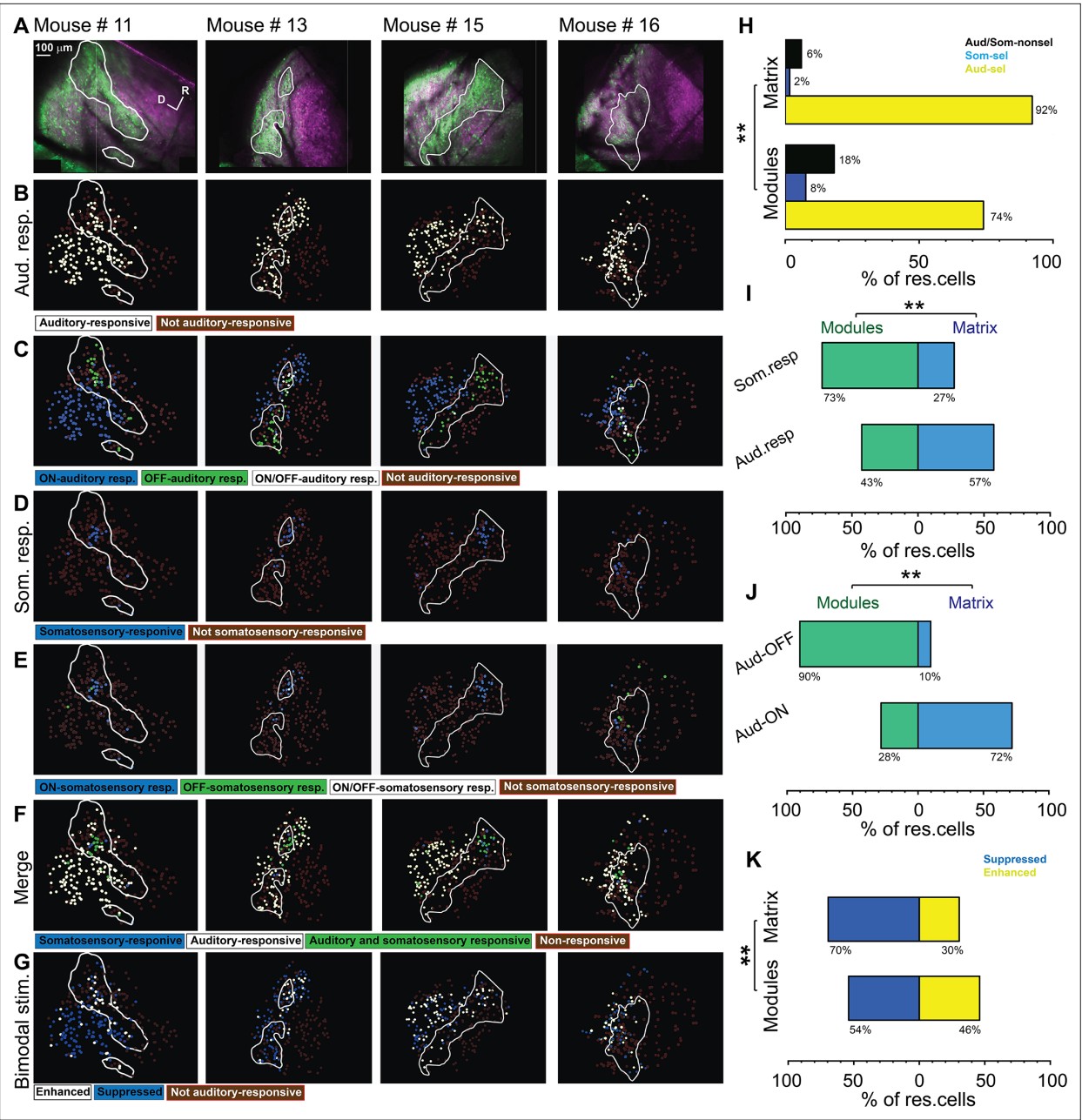

**Figure 7.** Types of somatosensory and auditory responses from the lateral cortex (LC) neurons imaged via microprism. (**A**) The two-photon (2P) images of GFP (green) and jRGECO1a expression (magenta) in the LC(microprism) show the modules within white irregular lines. (**B, D, F**) The pseudocolor images show the auditory (white circles) or somatosensory (blue circles) responsive cells of the LC(microprism) as well as the merged response (green circles), respectively. (**C, E**) The pseudocolor images show onset (ON) (blue circles), offset (OFF) (green circles), or onset/offset (ON/OFF) (white circles) responsive cells of the LC(microprism) to auditory and somatosensory stimuli, respectively. (**G**) The pseudocolor images show enhanced (white circles) and suppressive response (blue circles) of the auditory responsive cells of the LC(microprism) following the bimodal stimulation based on their response index (RI). (**H**) A bar graph showing the percentage of responsive cells to auditory stimulation only or auditory selective cells (Aud-sel, yellow bars), to somatosensory stimulation only or somatosensory selective cells (Som-sel, blue bars), or nonselectively to both auditory and somatosensory stimulations (Aud/Som-nonsel, black bars) in modules vs matrix (chi-square test, $\chi^2$=33.1, p=3.5 × 10$^{-6}$, % of responsive cells = 74%, 8%, and 18% for Aud-sel, Som-sel, and Aud/Som-nonsel, respectively at [modules, 235 cells] vs 92%, 2%, and 6% for Aud-sel, Som-sel, and Aud/Som-sel, respectively at [matrix, 297 cells] – 4 animals, **p<0.05 vs modules). (**I**) A bar graph showing the percentage of all cells having auditory responses (Aud. resp) or those having somatosensory responses (Som. resp) in modules vs matrix (chi-square test, $\chi^2$=26.0, p=9.3 × 10$^{-5}$, % of responsive cells [modules vs matrix]=43% vs 57% for [Aud-resp, 509 cells] and 73% vs 27% for [Som. resp, 84 cells] – 4 animals, **p<0.05 vs modules). (**J**) A bar graph showing the percentage of auditory responsive cells with onset (Aud-ON) vs those with offset (Aud-OFF) responses in modules vs matrix (chi-square test, $\chi^2$=120.5, p<0.001, % of responsive cells [modules vs matrix]=28% vs 72% for [Aud-ON, 390 cells] and 90% vs 10% for [Aud-OFF, 93 cells] – 4 animals, **p<0.05 vs modules). (**K**)

*Figure 7 continued on next page*

*Figure 7 continued*

A bar graph showing the percentage of auditory responsive cells with suppressed vs those with enhanced responses following bimodal stimulation in modules vs matrix (chi-square test, $\chi^2$=12.9, p=0.04, % of responsive cells = 54% vs 46% for suppressed vs enhanced [modules, 217 cells] and 70% vs 30% for suppressed vs enhanced [matrix, 292 cells] – 4 animals, **p<0.05 vs modules). D: dorsal, R: rostral.

auditory responses in both modules and matrix following bimodal stimulation. We observed that matrix and modules have more cells with suppressed auditory responses than those having enhanced auditory responses. However, matrix had a higher fraction of cells with suppressed responses than modules, while modules had a higher fraction of cells with enhanced responses (*Figure 7G and K*, chi-square test, $\chi^2$=12.9, p=0.04).

Consistent with the above data, the LC(dorsal surface) (*Figure 8A–D*, dorsolateral white rectangle) shared common response profiles with the LC(microprism). For instance, both regions had a higher percentage of somatosensory responsive cells than the DC (*Figure 8A–D*, dorsal yellow and dorsomedial blue rectangles) as quantitatively shown (*Figure 8H*, one-way ANOVA, p=0.02), which suggests that the LC is the main region of the IC shell that processes somatosensory information, consistent with the anatomical reports (*Aitkin et al., 1978*; *Jain and Shore, 2006*; *Lesicko Alexandria et al., 2016*; *Liu et al., 2023*). Similar to the LC(microprism), the matrix and modules imaged from the dorsal surface had more percentage of Aud-sel cells than Som-sel and Aud/Som-nonsel cells (*Figure 8E and I*, chi-square test, $\chi^2$=86.1, p<0.001). Collectively, the matrix had more auditory responsive cells than modules, while modules had more somatosensory responsive cells than matrix (*Figure 8J*, chi-square test, $\chi^2$=66.5, p<0.001). Additionally, more cells with onset auditory response were found in the matrix, while the cells with offset auditory responses were clustered in the modules (*Figure 8F and K*, chi-square test, $\chi^2$=500.2, p<0.001).

Consistent with the LC(microprism), some cells had suppressed or enhanced auditory responses following bimodal stimulation (*Figures 7K and 8L*). The matrix imaged from the dorsal surface also showed more cells with suppressed auditory responses than those having enhancing auditory responses induced by bimodal stimulation. Additionally, the matrix had a higher fraction of cells with suppressed responses than modules, while modules had a higher fraction of cells with enhanced responses than matrix. In contrast to the LC(microprism), the modules imaged directly from the dorsal surface had a higher fraction of cells with enhanced responses than those with suppressed responses, which may indicate that the ratio between suppressed and enhanced responses in modules could be altered based on the dorsoventral location (*Figure 8G and L*, chi-square test, $\chi^2$=18.5, p=1.6 × 10$^{-5}$). In general, these data confirmed the findings obtained from the LC imaged via microprism and emphasized the functional difference between the LC and DC.

## LC and DC differ in their acoustic and multimodal response properties in awake animals

To determine if the presence of ketamine/xylazine anesthesia influenced the response properties detailed above, additional experiments were performed in awake animals (n=6 total, 3 for the LC(microprism) and 3 for the DC). Similar to what was observed under anesthesia, the cells of the LC(microprism) did not show clear evidence of tonotopy (*Figure 9B*), which was indicated by a low R$^2$ of the regressions fit between the BTFs of the cells and their locations over different anatomical axes (*Figure 9—figure supplement 1A*). Similarly, the cells of the modules were tuned to relatively higher frequencies than those in matrix (Mann-Whitney test, z=3.8, p=1.0 × 10$^{-4}$, median of best frequency = 10 vs 7.1 kHz for modules [148 cells from 3 animals] vs matrix [287 cells from 3 animals]). Additionally, the data obtained from imaging the DC under awake preparations showed a similar tonotopic map (*Figure 9D*) as that observed under anesthesia indicated by a high R$^2$ of the regression fit between the BTFs of the cells and their locations over the rostromedial and caudolateral axis (*Figure 9—figure supplement 1B*). Yet, a greater proportion of high-frequency cells were visually observed in the mediodorsal portion of the DC, which was indicated by a higher R$^2$ of the nonlinear quadratic fit than that of the linear fit, leading to the observation of a narrow low-frequency zone similar to the previous work under awake preparations (*Wong and Borst, 2019*). The LC(microprism) had also more cells tuned to higher frequencies compared to the DC (*Figure 10A*, Mann-Whitney test, p=1.6 × 10$^{-6}$). We also examined the width of the receptive field (RF) of the DC vs LC neurons at each tested sound level under anesthetized and awake preparations using the binarized receptive field sum (RFS) method (*Bowen et al., 2020*). Both

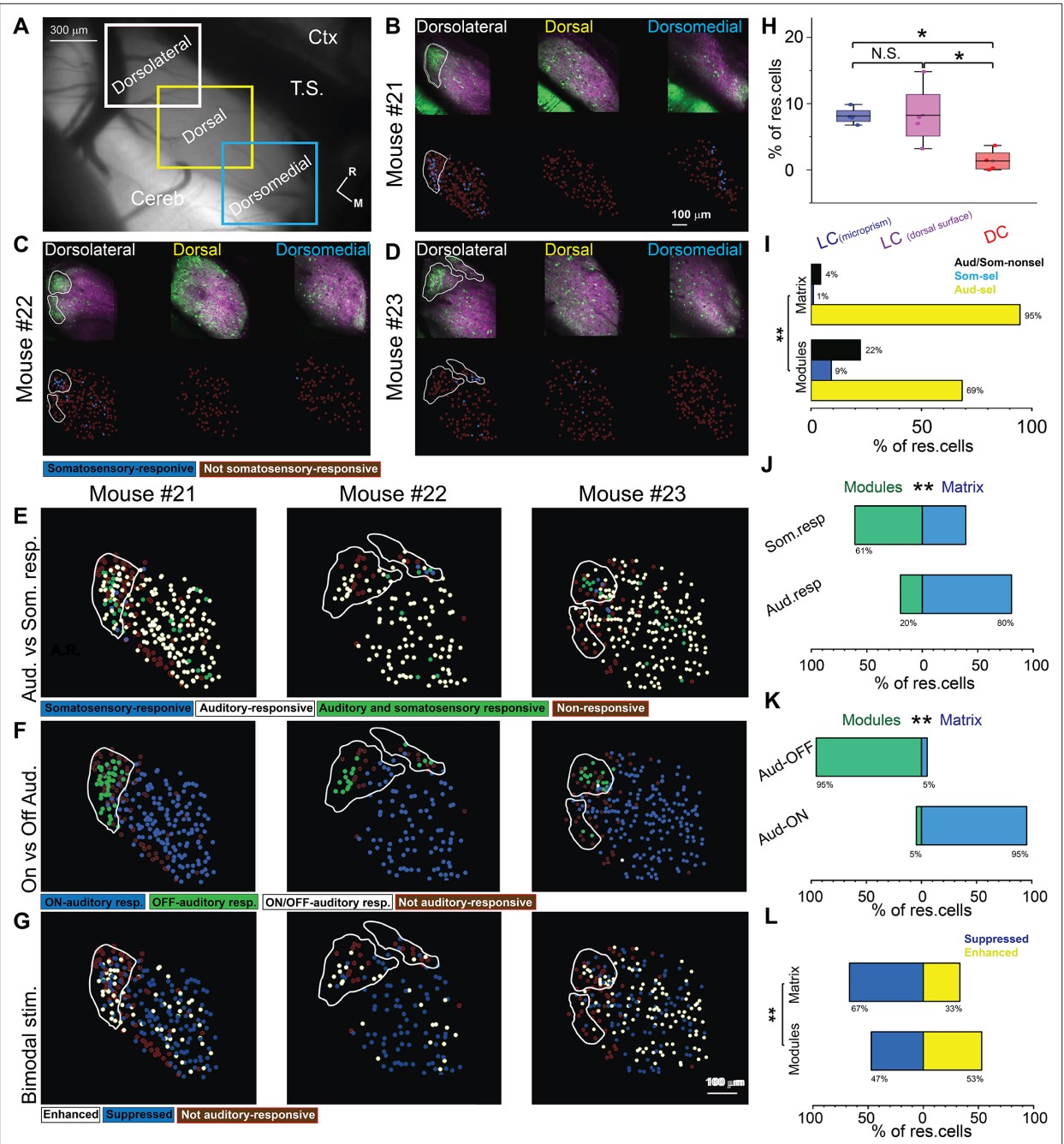

**Figure 8.** The somatosensory vs auditory responses over the dorsal cortex (DC) neurons. (**A**) A low magnification of the image obtained from the inferior colliculus (IC) surface. (**B–D**) The top row: the two-photon (2P) images of GFP (green) and jRGECO1a expression (magenta) of three main regions (dorsolateral, dorsal, and dorsomedial) of the DC surface imaged from the dorsal surface showing the modules within irregular white lines in the dorsolateral region. The bottom row: the pseudocolor images show the somatosensory responsive cells (blue circles) after somatosensory stimulations. (**E**) The pseudocolor images show the responsive cells of the LC(dorsal surface) to auditory (white circles), somatosensory (blue circles), or both stimulations (green circles). (**F**) The pseudocolor images show auditory onset (Aud-ON – blue circles), offset (Aud-OFF, green circles), or onset/offset (Aud-ON/ OFF, white circles) responsive cells of the LC imaged from the dorsal surface to auditory stimulation. (**G**) The pseudocolor images show the enhanced (white circles) and suppressive (blue circles) response of the auditory responsive cells of the LC imaged from the dorsal surface following the bimodal stimulation based on their response index (RI). (**H**) A box plot shows the percentage of somatosensory responsive cells in the LC(microprism), the LC(dorsal surface), or the DC (one-way ANOVA, p=0.02, the percentage of somatosensory responsive cells mean ± SEM = 1.3 ± 0.83% [DC, 4 animals, 8.2 ± 2.4%] [LC(dorsal surface), 4 animals], and 8.1 ± 0.64% [LC(microprism), 4 animals], *p<0.05, vs DC). (**I**) A bar graph showing the percentage of responsive cells to auditory stimulation only or auditory selective cells (Aud-sel, yellow bars), to somatosensory stimulation only or somatosensory selective cells (Som-sel, blue bars), or nonselectively to both auditory and somatosensory stimulations (Aud/Som-nonsel, black bars) in modules vs matrix (chi-square test, $\chi^2$=86.1,

*Figure 8 continued on next page*

*Figure 8 continued*

p<0.001, % of responsive cells = 69%, 9%, and 22% for Aud-sel, Som-sel, and Aud/Som-nonsel [modules, 152 cells] and 95%, 1%, and 4% for Aud-sel, Som-sel, and Aud/Som-nonsel [matrix, 573 cell] – 4 animals, **p<0.05 vs modules). (**J**) A bar graph showing the percentage of all cells having auditory responses (Aud. resp) or those having somatosensory responses (Som. resp) in modules vs matrix (chi-square test, $\chi^2$=66.5, p<0.001, % of responsive cells [modules vs matrix] = 20% vs 80% for [Aud. resp, 705 cells] and 61% vs 39% for [Som. resp, 79 cells] – 4 animals, **p<0.05 vs modules). (**K**) A bar graph showing the percentage of auditory responsive cells with onset (Aud-ON) vs those with offset (Aud-OFF) responses in modules vs matrix (chi-square test, $\chi^2$=500.2, p<0.001, % of responsive cells [modules vs matrix]=5% vs 95% for [Aud-ON, 584 cells] and 95% vs 5% for [Aud-OFF, 116 cells] – 4 animals, **p<0.05 vs modules). (**L**) A bar graph showing the percentage of auditory responsive cells with suppressed vs those with enhanced responses following bimodal stimulation in modules vs matrix (chi-square test, $\chi^2$=18.5, p=1.6 × $10^{-5}$, % of responsive cells = 47% vs 53% for suppressed vs enhanced [modules, 138 cells] and 67% vs 33% for suppressed vs enhanced [matrix, 567 cells] – 4 animals, **p<0.05 vs modules). Cereb: cerebellum, Ctx: cerebral cortex, M: medial, R: rostral, TS: transverse sinus.

DC and LC of the anesthetized animals had higher RFS mean and median than those of awake animals given that ketamine was known to broaden the RF (***Guo et al., 2012***). In both preparations (anesthetized and awake), the DC had a higher RFS mean than that of the LC (***Figure 10—figure supplement 1***, Kruskal-Wallis ANOVA, p<0.001), which could be consistent with the finding that the DC had a lower SMI than LC.

Similar to what was shown in the anesthetized animals, the fraction of Noise-sel cells was higher than that of Tone-sel cells in both LC(micrprism) and DC imaged directly from the top in awake animals (***Figure 10B and D***). In contrast to the anesthetized animals, the fraction of Non-sel cells had different profiles. For instance, while the percentage of Non-sel cells was higher than Noise-sel and Tone-sel cells in the LC(microprism) (***Figure 10B***, one-way ANOVA, $f_{(2,6)}$ = 41.6, p=3.0 × $10^{-4}$), they were significantly lower than that of Noise-sel cells and similar to that of Tone-sel cells in the DC (***Figure 10D***, one-way ANOVA, $f_{(2,6)}$ = 12.8, p=0.006). Despite that modules and matrix had similar fractions of Noise-sel, Tone-sel, and Non-sel cells (***Figure 10C***, Mann-Whitney test [modules vs matrix], z=–0.46, –0.46, and –1.85, p=0.64, 0.64, and 0.06 for Noise-sel, Tone-sel, and Non-sel, respectively), matrix had more spectrally (***Figure 10E***, two-way ANOVA: p=5.3 × $10^{-10}$, 1.0, and 0.17 for the region, sound level, and interaction, respectively) and temporally (***Figure 10F***, two-way ANOVA: p=4.6 × $10^{-7}$, 1.0, and 0.11 for the region, sound level, and interaction, respectively) modulated cells than modules, which was similar to anesthetized animals. Additionally, the LC(microprism) and DC showed different profiles than those found by anesthetized animals. While LC and DC had similar SMI (***Figure 10G***, two-way ANOVA: p=0.22, 2.1×$10^{-12}$, and 1.1×$10^{-6}$ for the region, sound level, and interaction, respectively), the LC(microprism) had a lower average of TMIs compared to the DC (***Figure 10H***, two-way ANOVA: p=1.9 × $10^{-9}$, 6.6×$10^{-24}$, and 0.01 for the region, sound level, and interaction, respectively).

Similar to the anesthetized preparations, the auditory and somatosensory responsive cells were topographically organized across modules and matrix (***Figure 11A, C, and E***). However, it was noticed that the somatosensory and auditory responsive cells were clustered within a pool of nonresponsive cells (***Figure 11E***). Some of these nonresponsive cells to unmodulated broadband noise (***Figure 11—figure supplement 1A***) were found to respond to pure tones (***Figure 11—figure supplement 1B***, blue arrows), indicating that these nonresponsive cells may be sensitive to a specific feature or type of sensory stimulations. Additionally, some of the cells that did not respond to any of the tested stimulations showed spontaneous activity (***Figure 11—figure supplement 1C***), indicating the viability of the nonresponsive cells. Within the pool of responsive cells, the matrix and modules imaged via microprism had higher fractions of Aud-sel cells than those of Som-sel and Aud/Som-nonsel cells (***Figure 11E and G***, chi-square test, $\chi^2$=42.5, p=4.4 × $10^{-8}$), which was similar to the data obtained from the anesthetized animals. Additionally, matrix had a higher fraction of auditory responsive cells than modules, while modules had a higher fraction of somatosensory responsive cells than matrix (***Figure 11A, D, and H***, chi-square test, $\chi^2$=30.2, p=1.2 × $10^{-6}$). In contrast to the anesthetized animals, the offset cells were very rare (only 6 cells/3 animals), and were evenly distributed between modules and matrix resulting in no significance difference between matrix and modules (***Figure 11C and I***, chi-square test, $\chi^2$=2.8, p=0.41). Similar to anesthetized data, we found that matrix and modules had higher fractions of cells with auditory suppressed responses than those with auditory enhanced responses induced by bimodal stimulation, indicating an overall suppression in the cellular auditory response. In contrast to the anesthetized animals, modules had a higher fraction of cells with suppressed response than matrix and a lower fraction of cells with enhanced responses compared to matrix (***Figure 11G and J***, $\chi^2$

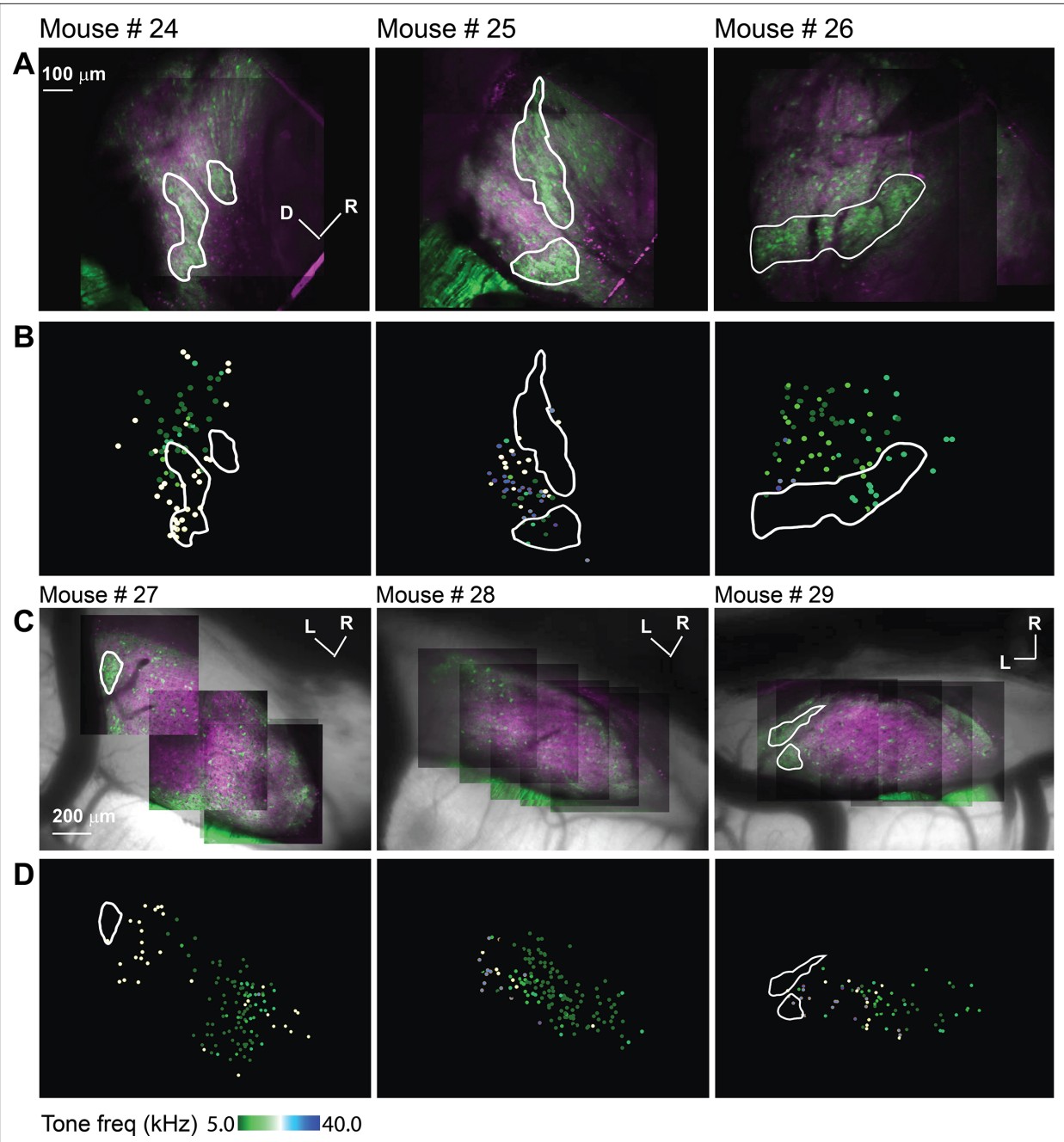

**Figure 9.** The acoustic response from LC$_{(microprism)}$ and dorsal cortex (DC) imaged from the dorsal surface in awake animals. (**A, C**) The two-photon (2P) images of GFP (green) and jRGECO1a expression (magenta) on either the LC$_{(microprism)}$ surface imaged via the microprism or the DC imaged directly from the dorsal surface, respectively showing the modules within irregular white lines. (**B, D**) The pseudocolor images show the responsive cells to the pure tone of different combinations of frequencies and levels of the LC$_{(microprism)}$ or the DC imaged directly from the dorsal surface, respectively. D: dorsal, L: lateral, R: rostral.

The online version of this article includes the following figure supplement(s) for figure 9:

**Figure supplement 1.** The best regression fit between best tone freuency and different anatomical axes.

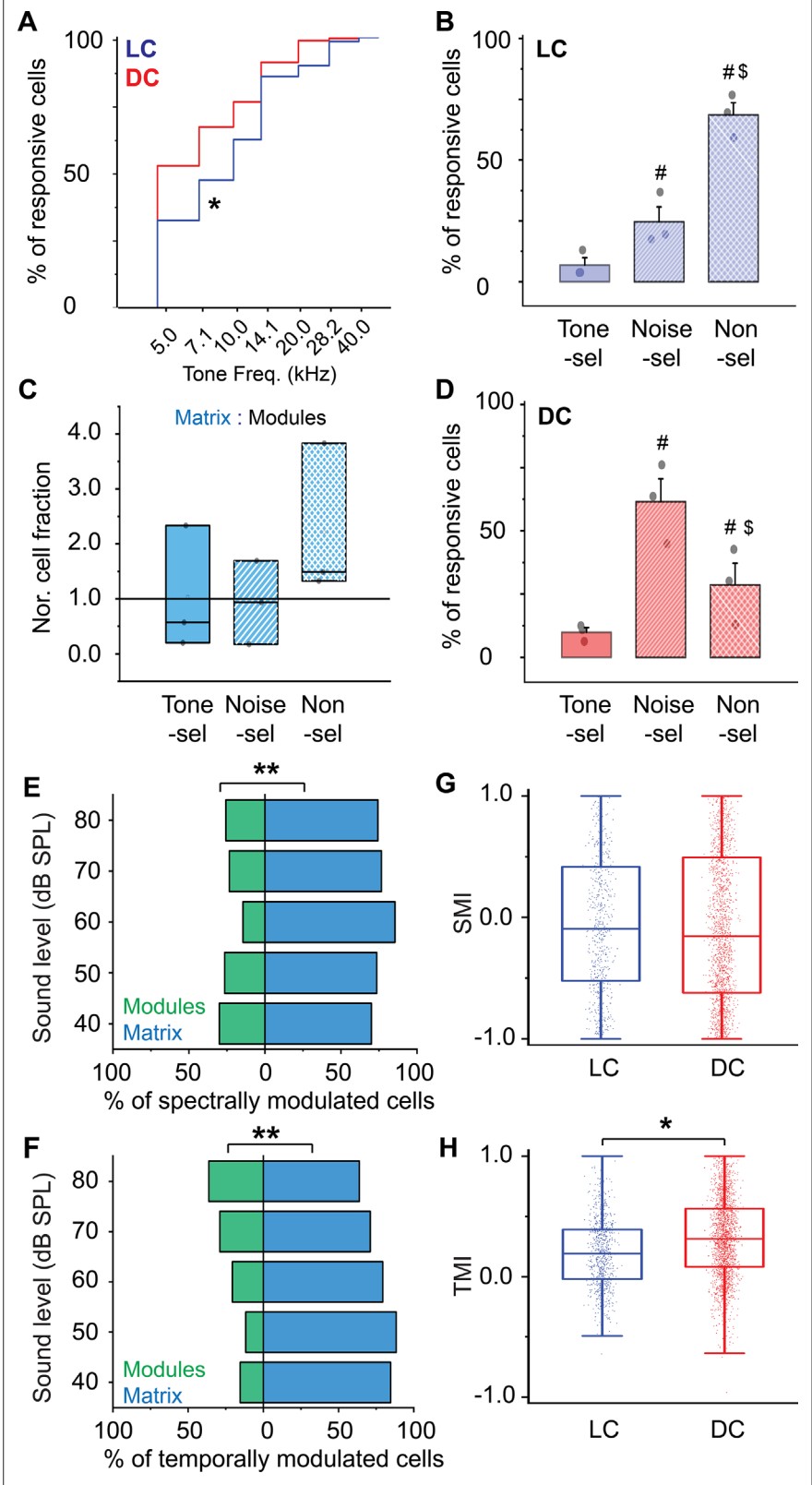

**Figure 10.** The cellular response to pure tones and AM-noise of the LC(microprism) vs dorsal cortex (DC) imaged from the dorsal surface in awake animals. (**A**) The cumulative distribution function of the best tone frequencies of all cells collected either from the lateral cortex (LC) imaged via microprism (blue line) or the DC imaged directly from its dorsal surface (red line) (Mann-Whitney test, z=–4.8, p=1.6 × 10⁻⁶, a median of best-tuned frequency

*Figure 10 continued on next page*

*Figure 10 continued*

= 5 and 10 kHz for the DC [767 cells from 3 animals] and the LC$_{(microprism)}$ [371 cells from 3 animals], respectively, *p<0.05 vs DC). (**B**) A bar graph showing the fractions of cells responding to only pure tone (Tone-sel), only AM-noise (Noise-sel), or to both nonselectively (Nonsel) on the LC$_{(microprism)}$ (3 animals) (one-way ANOVA, f$_{(2,6)}$ = 41.6, p=3.0 × 10$^{-4}$, Fisher's post hoc test: p=0.04 and 1.1×10$^{-4}$ for Noise-sel and Non-sel vs Tone-sel, respectively, p=7.3 × 10$^{-4}$ for Noise-sel vs Non-sel, % of responsive cells ± SEM = 6 ± 3%, 24 ± 6%, and 68 ± 5% for Tone-sel, Noise-sel, and Non-sel, respectively, #p<0.05 vs Tone-sel and $p<0.05 vs Noise-sel). (**C**) Box graphs showing the fractions of Tone-sel, Noise-sel, and Non-sel cells within modules and matrix of the LC$_{(microprism)}$. (**D**) A bar graph showing the fractions of Tone-sel, Noise-sel, or Non-sel cells on the DC imaged directly from the dorsal surface (3 animals) (one-way ANOVA, f$_{(2,6)}$ = 12.8, p=0.006, Fisher's post hoc test: p=0.002 and 0.12 for Noise-sel and Non-sel vs Tone-sel, respectively, p=0.01 for Noise-sel vs Non-sel, % of responsive cells ± SEM = 9 ± 1%, 61 ± 9%, and 28 ± 8% for Tone-sel, Noise-sel, and Non-sel, respectively, #p<0.05 vs Tone-sel and $p<0.05 vs Noise-sel). (**E, F**) Bar graphs showing the percentage of spectrally (two-way ANOVA: f$_{(1,4,16)}$ = 173.0, 0, and 1.8 p=5.3 × 10$^{-10}$, 1.0, and 0.17 for the region, sound level, and interaction, respectively, % of spectrally modulated cells ± SEM = 23 ± 2% vs 76 ± 2% for modules vs matrix – 3 animals, **p<0.05 vs modules) or temporally (two-way ANOVA: f$_{(1,4,16)}$ = 65.9, 0, and 2.2 – p=4.6 × 10$^{-7}$, 1.0, and 0.11 for the region, sound level, and interaction, respectively, % of temporally modulated cells ± SEM = 23 ± 2% vs 76 ± 2% for modules vs matrix, n=3 animals, **p<0.05 vs modules) modulated cells, respectively, across different sound levels in modules (green bars) vs matrix (blue bars) imaged via microprism (**p<0.05, matrix vs modules). (**G, H**) Box plots showing the mean (black lines) and the distribution (colored dots) of the SMI (two-way ANOVA: f$_{(1,4,2411)}$=1.4, 15.3, and 8.3 – p=0.22, 2.1×10$^{-12}$, and 1.1×10$^{-6}$ for the region, sound level, and interaction, respectively, SMI ± SEM = –0.04±0.02 to –0.06 ± 0.01 for LC$_{(microprism)}$ vs DC – n=619 cells from 3 animals [LC$_{(microprism)}$], and 1826 cells from 3 animals [DC]) and TMI (two-way ANOVA: f$_{(1,4,2869)}$=36.2, 29.3, and 3.0 – p=1.9 × 10$^{-9}$, 6.6×10$^{-24}$, and 0.01 for the region, sound level, and interaction, respectively, SMI ± SEM = 0.2 ± 0.01 vs 0.32 ± 0.007 for LC$_{(microprism)}$ vs DC – n=781 cells from 3 animals (LC$_{(microprism)}$), and 2098 cells from 3 animals (DC), *p<0.05 vs DC). DC: dorsal cortex, LC: lateral cortex, SMI: spectral modulation index, TMI: temporal modulation index.

The online version of this article includes the following figure supplement(s) for figure 10:

**Figure supplement 1.** The receptive field sum (RFS) of dorsal cortex (DC) and lateral cortex (LC) neurons at different experimental preparations.

---

test = 31.9, p=5.2 × 10$^{-7}$), indicating the possibility that the bimodal interactions in modules could be affected by the arousal state of the animal.

## Discussion

### Summary

As a new imaging approach, a microprism was used to image the LC surface, to provide the first-ever sagittal image of the LC in vivo and to examine its functional organization, which revealed the distinction between strategies used by LC and DC for processing acoustic and non-acoustic information. Imaging the LC either via microprism or from the dorsal surface revealed that it contains GABAergic modules that were embedded in a non-GABAergic matrix. Although modules and matrix were found to be activated by somatosensory and auditory stimuli, they were functionally distinct. The matrix had more auditory-responsive cells with a lower threshold for complex sound detection and had more cells responsive to complex spectral and temporal auditory features than modules. In contrast, modules had more somatosensory responsive cells, consistent with anatomical differences in input to the modules compared to matrix (*Lesicko Alexandria et al., 2016*). Somatosensory stimulation suppressed auditory responses in the matrix, suggesting that multisensory interactions exist between the matrix and modules.

### The novelty of the imaging setup and technical considerations

Microprism-based 2P microscopy has been successfully used to image the cortical layers in the somatosensory and visual cortices of mice (*Andermann et al., 2013*) and to image other inaccessible cortical surfaces like those folded in the brain sulci such as the medial prefrontal cortex (*Low et al., 2014*) or entorhinal cortex (*Beckmann et al., 2019*; *Heys et al., 2014*; *Low et al., 2014*). In those studies, the glass window glued to the microprism was used, which requires significant skill and potential removal of brain regions to allow prism access (*Heys et al., 2014*). Another approach is to use a micro metal or

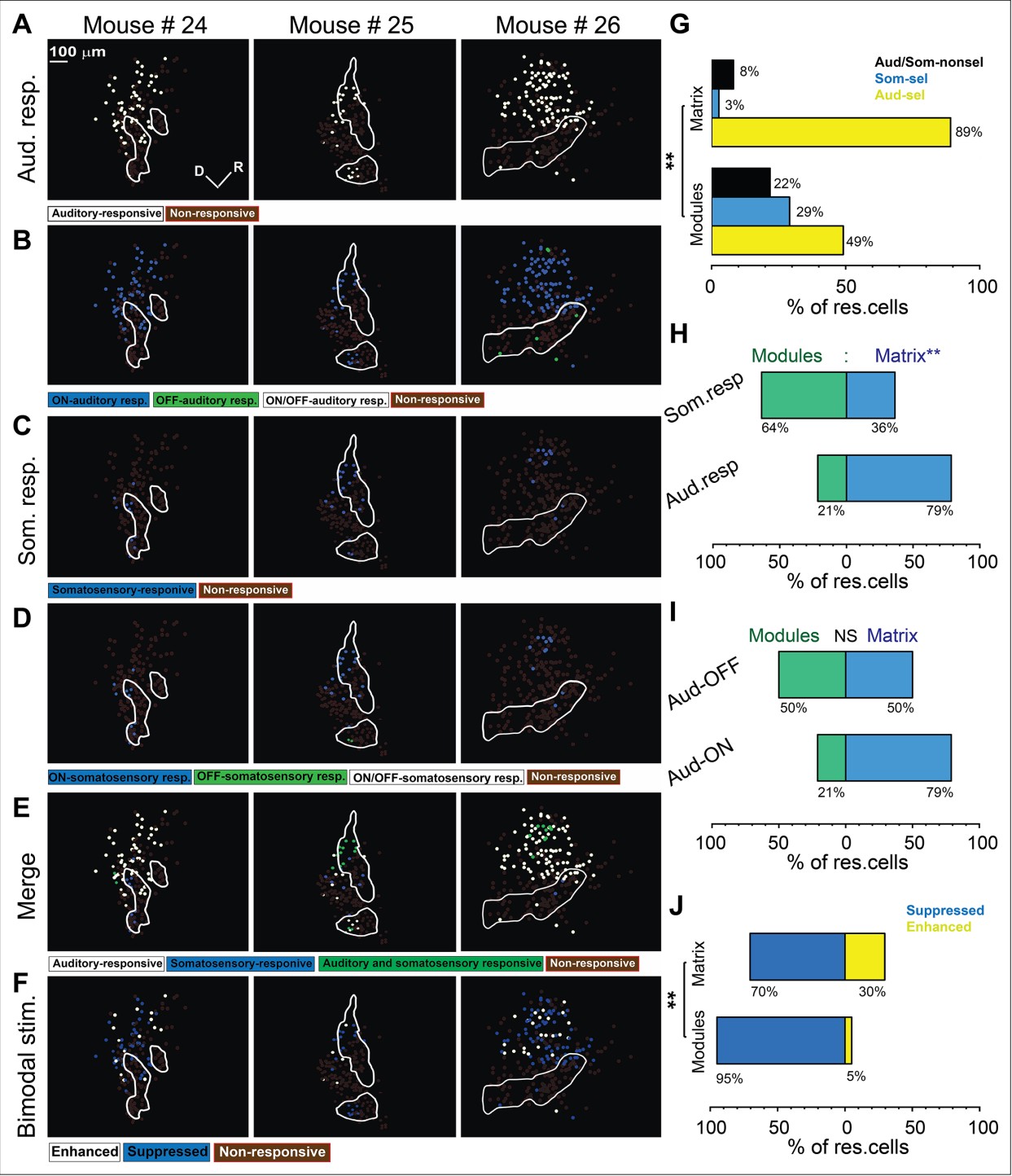

**Figure 11.** The somatosensory responses from LC$_{(microprism)}$ in awake animals. (**A, C, E**) The pseudocolor images show the auditory (white circles) or somatosensory (blue circles) responsive cells of the LC$_{(microprism)}$ as well as the merged response (green circles), respectively. (**B, D**) The pseudocolor images show onset (ON) (blue circles), offset (OFF) (green circles), or onset/offset (ON/OFF) (white circles) responsive cells of the LC$_{(microprism)}$ to auditory and somatosensory stimuli, respectively. (**F**) The pseudocolor images show enhanced (white circles) and suppressive response (blue circles) of the auditory responsive cells of the LC$_{(microprism)}$ following the bimodal stimulation based on their response index (RI). (**G**) A bar graph showing the percentage of responsive cells to auditory stimulation only or auditory selective cells (Aud-sel, yellow bars), to somatosensory stimulation only or somatosensory selective cells (Som-sel, blue bars), or nonselectively to both auditory and somatosensory stimulations (Aud/Som-nonsel, black bars) in modules vs matrix (chi-square test, $\chi^2$=42.5, p=4.4 × 10$^{-8}$, % of responsive cells = 49%, 29%, and 22% for Aud-sel, Som-sel, and Aud/Som-nonsel [modules, 55 cells] and 89%, 3%, and 8% for Aud-sel, Som-sel, and Aud/Som-nonsel [matrix, 147 cells] – 3 animals, **p<0.05 vs modules). (**H**) A bar graph showing the

*Figure 11 continued on next page*

*Figure 11 continued*

percentage of all cells having auditory responses (Aud. resp) or those having somatosensory responses (Som. resp) in modules vs matrix (chi-square test, $\chi^2$=30.2, p=1.2 × 10$^{-6}$, % of responsive cells [modules vs matrix]=21% vs 79% for [Aud-resp, 182 cells], 64% vs 36% for [Som-resp, 44 cells] – 3 animals, **p<0.05 vs modules). (**I**) A bar graph showing the percentage of auditory responsive cells with onset (Aud-ON) vs those with offset (Aud-OFF) responses in modules vs matrix (chi-square test, $\chi^2$=2.8, p=0.41, % of responsive cells [modules vs matrix]=21% vs 79% for [Aud-ON, 176 cells], 50% vs 50% for [Aud-OFF, 6 cells] – 3 animals, **p<0.05 vs modules). (**J**) A bar graph showing the percentage of auditory responsive cells with suppressed vs those with enhanced responses following bimodal stimulation in modules vs matrix (chi-square test = 31.9, p=5.2 × 10$^{-7}$, matrix: the percentage of responding cells = 95% vs 5% for suppressed vs enhanced [modules, 40 cells] and 70% vs 30% for suppressed vs enhanced [matrix, 142 cells] – 3 animals, **p<0.05 vs modules). D: dorsal, R: rostral.

The online version of this article includes the following figure supplement(s) for figure 11:

**Figure supplement 1.** The activity of the nonresponsive cells of the LC$_{(microprism)}$ in awake animals.

diamond-made knife to create a pocket with minimal tissue damage (*Andermann et al., 2013*; *Marks and Goard, 2021*). Accordingly, we combined and modified some of these procedures to improve the microprism implantation to image the sagittal surface of the LC, which is deeply embedded into the brain tissue in a highly vascular area, with minimum damage to the IC or the surrounding vasculature.

Previous work using microprisms has been done chronically for up to 60 days (*Heys et al., 2014*; *Low et al., 2014*; *Wenzel et al., 2017*), with minimal tissue inflammation (*Andermann et al., 2013*). The current study was initially done on anesthetized animals to examine and validate the new approach, which later revealed similar organizational differences between LC and DC in awake animals. Although imaging through a microprism and 1 mm cover glass appears to have minimal impact on image quality compared to direct 2P imaging (*Chia and Levene, 2009*), it was important to compare LC imaging via microprism with the direct 2P imaging of the dorsal surface to validate the new imaging method. We found no significant differences between LC response properties when imaging via microprism or imaging directly from the dorsal surface. However, we did observe significant functional differences between the LC and the DC.

## Responsiveness to spectral and temporal features

While the LC showed inconsistent and fragmented tonotopy, the DC showed a consistent one, as previously reported (*Barnstedt et al., 2015*; *Wong and Borst, 2019*). These data were supported by a lower $R^2$ of the best regression fit between BTFs of the LC cells and their locations compared to a higher $R^2$ obtained for the DC. The direction of the best regression fit shown by the LC was inconsistent across animals, which could be related to the fact that the microprism did not permit a full view of the LC surface. Given that microprism permits imaging of only a single area per animal, it is expected that each animal may have a different profile of cellular organization based on the location of the imaged area on the LC.

The surface of the DC was biased to lower frequencies compared to LC, consistent with previous imaging and recording studies (*Ito et al., 2014*; *Romand and Ehret, 1990*; *Shen et al., 2003*; *Stiebler and Ehret, 1985*). Despite their response to pure tones, the DC and LC, particularly in the matrix, were more responsive to AM-noise, which fits their function as non-lemniscal divisions (*Chen et al., 2018*; *Faye Lund and Osen, 1985*; *Loftus et al., 2008*).

The spectral and temporal features of animal vocalization and speech are not independent (*Singh and Theunissen, 2003*). We observed that LC neurons, particularly in the matrix, had greater responsiveness to spectral and temporal modulation than the DC. Given the role of inhibition in spectral and temporal processing (*Cai et al., 2018*; *Chang et al., 2005*; *Wehr and Zador, 2003*), GABAergic cells of the modules could play an important role in increasing the SMI and TMI, especially at lower sound levels, which could be critical for localizing objects in noisy environments given the importance of spectral and temporal cues for sound localization (*Davis et al., 2003*; *Goodman et al., 2013*; *Sinex, 2005*). Therefore, future work is critical to image the functional responses of the GABAergic cells of the modules and compare them with the non-GABAergic cells inside and outside the modules.

## Somatosensory vs auditory responses

Consistent with the previous anatomical findings (*Lesicko Alexandria et al., 2016*), modules showed more somatosensory responsive cells than matrix. Since modules receive somatosensory inputs from different brain regions (*Lesicko Alexandria et al., 2016*; *Lohse et al., 2022*; *Zhou and Shore, 2006*),

the somatosensory response shown by modules could be modulated by different types of somatosensory stimulations, which is an open area for future studies. In contrast to modules, we observed that most auditory-responsive neurons, particularly onset-responsive neurons, were found in the matrix, which receives mostly auditory inputs (*Lesicko Alexandria et al., 2016*). However, a subpopulation of acoustically driven offset-responsive neurons preferably resides in the modules compared to the matrix. Given that the offset response could be generated as a rebound to the termination of the ongoing inhibition during the sound presentation (*Xu et al., 2014*), GABAergic cells in the modules could play an important role in the coding of sound termination as found before in the cortex (*Solyga and Barkat, 2021*). Because some of the offset acoustically driven cells inside the modules were also activated by the somatosensory stimulation, this fraction of cells could then encode the termination of somatosensory-associated sounds through the local inhibitory circuits of the modules. Under in vivo recording of the mouse IC, the offset responses were previously reported from cells that were non-GABAergic and located in the caudal and shallower parts of the IC (*Kasai et al., 2012*). These data suggest that the module/matrix system permits the preservation of distinct multimodal response properties, which is consistent with their anatomical distinction, despite the massive integration of inputs in the LC.

Consistent with previous reports (*Aitkin et al., 1978*; *Aitkin et al., 1981*; *Jain and Shore, 2006*), we observed that combined auditory-somatosensory stimulation generally suppressed neural responses to auditory stimuli and that this suppression was the most prominent in the LC matrix of anesthetized animals. Mechanistically, it is not known if such suppressive responses are created in the LC or are inherited from other brain regions since auditory-somatosensory interactions occur as early as in the cochlear nucleus, and are also found in the IC, thalamus, and cortex (*Aitkin et al., 1981*; *Haenggeli et al., 2005*; *Kimura et al., 2007*; *Li and Mizuno, 1997*; *Lohse et al., 2021*; *Shore, 2005*; *Shore et al., 2000*; *Wong et al., 2015*), reviewed in *Brunelle and Llano, 2023*; *Lohse et al., 2022*. We note that a systematic exploration of the stimulus space was not conducted here such that other acoustic or somatosensory stimuli, different relative amplitudes, or timings could have produced enhancement rather than suppression. Future work will be needed to further explore this stimulus space to more fully understand multisensory processing in the LC.

## Effect of ketamine anesthesia

The data obtained from anesthetized animals were similar to those obtained from awake animals in the findings related to the tonotopy, the preference of the cells to respond to AM-noise in the LC and DC, the increased sensitivity of matrix to detect the spectral and temporal features of sound, and the bimodal response properties of the matrix vs modules. However, anesthetized and awake animals showed some differences that could be related to the effect of ketamine. For instance, ketamine was found to broaden the RF of the auditory-responsive neurons (*Guo et al., 2012*). Consistently, the width of the RF of all DC and LC neurons was higher under ketamine anesthesia compared to awake preparation. However, the DC neurons had a wider RF than those in the LC. The relatively narrower RF of the LC could be consistent with their higher SMI (discussed below) compared to that of the DC. Both findings could be related to the degree of inhibition that may shape the inhibitory bands of the RF in the LC neurons given that the LC has prominent GABAergic modules. Also, while the average of the SMIs was negative with no difference between the LC$_{(microprism)}$ and the DC of awake animals, it was positive with a higher value in the LC$_{(microprism)}$ compared to the DC of the anesthetized animals, indicating the preference of cells to respond more to pure tones under anesthesia. Given that ketamine was reported to increase the duration of the tone-evoked response in the auditory cortex (*Guo et al., 2012*), the same effect could increase the area under the curve (AUC) for the tone-evoked response that might result in a positive average of SMI. Additionally, while LC had a higher average of TMIs than that of the DC of anesthetized animals, the LC had a lower mean of TMIs than that of the DC of awake animals, indicating that ketamine may specifically modulate the response properties of the LC via its effects on GABAergic cells (*Behrens et al., 2007*; *Deane et al., 2020*; *Weckmann et al., 2019*), given that GABAergic inhibition reportedly contribute to the processing of the temporally modulated signals (*Alluri et al., 2021*; *Burger and Pollak, 1998*; *Cai et al., 2018*; *Gourévitch et al., 2020*). In both anesthetized and awake animals, matrix neurons always had a consistent ratio between the cells with suppressed (~70%) and enhanced (~30%) auditory responses induced by bimodal stimulation. However, modules had variable ratios across anesthetized and awake animals. While modules had

a lower fraction of cells with enhanced responses than matrix in awake animals, they had a higher fraction of cells with enhanced responses in anesthetized animals, which could be consistent with the direct inhibitory effect of ketamine on GABAergic cells (*Behrens et al., 2007*; *Deane et al., 2020*; *Weckmann et al., 2019*). Given that modules are clusters of GABAergic cells and terminals, modules could be specifically modulated by ketamine anesthesia.

## Conclusions

Here, using a novel microprism approach to image the LC in anesthetized and awake mice, we observed that (1) DC displayed tonotopic organization which was not consistently present in the LC, (2) DC neurons were generally more sound-responsive than LC neurons but that neurons in both regions had preferential responses to spectrally and temporally complex stimuli, (3) matrix regions of the LC were more responsive to spectrally and temporally complex stimuli than modules and that modules were more responsive to somatosensory stimuli, and (4) bimodal stimulation led to suppressive responses that were more prominent in the matrix. In general, these data highlight significant organizational differences between the LC and DC and between modules and matrix in the LC. Future work using microprisms in the LC will help to reveal the potential differential role of matrix vs module neurons in multisensory integration tasks in behaving animals.

# Materials and methods

**Key resources table**

| Reagent type (species) or resource | Designation | Source or reference | Identifiers | Additional information |
| --- | --- | --- | --- | --- |
| Strain, strain background (*Mus musculus*) | GAD67-GFP (GAD1GFP) | Generously given | N/A | Developed and shared with permission from Dr. Yuchio Yanagawa at Gunma University and obtained from Dr. Douglas Oliver at the University of Connecticut |
| Strain, strain background (*Mus musculus*) | Tg(Thy1-jRGECO1a) GP8.20Dkim/J | Jackson Laboratory, USA | 30525 | |
| Sequence-based reagent | jRGECO1a – primer | Transnetyx, USA | mApple-1 Tg | |
| Antibody | NeuN (D4G40) XP Rabbit mAB | Cell Signaling, USA | 62994 | (1:100) |
| Antibody | Goat anti-rabbit IgG conjugated to Alexa Fluor 405 | Thermo Fisher, USA | A-31556 | (1:200) |

## Animal subjects

Male mice of 8–12 weeks of age were used. Given that our preliminary data suggest male/female differences in DC organization (*Ibrahim et al., 2022*) which will be published separately, the male mice were only used here. GAD67-GFP (*GAD1GFP*) knock-in mice (developed and shared with permission from Dr. Yuchio Yanagawa at Gunma University and obtained from Dr. Douglas Oliver at the University of Connecticut), where GFP is exclusively expressed in GABAergic cells (*Tamamaki et al., 2003*), were used to visualize the GABAergic cells and LC modules. To simultaneously monitor calcium signals and visualize the GFP+ cells, Tg(*Thy1*-jRGECO1a)GP8.20Dkim/J mice (Jackson Laboratory, Stock# 030525) were crossed with GAD67-GFP knock-in mice to generate GAD67-GFPx*Thy1*-jRGECO1a hybrid mice. The mice were housed and bred at the animal research facility of The Beckman Institute for Advanced Science and Technology. The animal care facilities are approved by the Association for Assessment and Accreditation of Laboratory Animal Care (AAALAC). To phenotype the GFP-positive animals, a single-photon fluorescence microscope was used for the transcranial examination of the pups at postnatal day 4 using excitation: 472/30 nm and emission 520/35 nm filters along with dichroic 505 nm long pass at low power magnification (×2.5/0.08, Olympus Objectives, MPlanFL N, Japan). The positive pups exhibited a green fluorescence in the cerebellum, the cerebral cortex, and the olfactory bulb and were only kept with their parents until weaning. The genotyping for jRGECO1a was done using a probe composed of a forward primer: GCCGCCGAGGTCAAGA and a reverse primer: TCCAACTTGATGTCGACGATGTAG by Transnetyx (Transnetyx, USA). Samples of the mice's tail snips around 1 month of age were used for genotyping. All applicable guidelines for the care and

use of animals were followed. All procedures were approved by the Institutional Animal Care and Use Committee (IACUC).

## Surgery

The general procedures of craniotomy were previously described (*Goldey et al., 2014*), with modifications to place craniotomy over the IC. Before surgery, mice were anesthetized with a mixture of ketamine, xylazine, and acepromazine (100, 3, and 3 mg/kg, respectively) delivered intraperitoneally. The anesthesia was maintained during the surgery and imaging using only ketamine (100 mg/kg). To prevent neural edema during or after the craniotomy, an intramuscular injection of dexamethasone sodium (4.8 mg/kg) was given just before the surgery using an insulin syringe. After placing the animal in the stereotaxic apparatus (David Kopf Instruments, USA), both eyes were protected by applying Optixcare lubricant eye gel (Aventix Animal Health, Canada). The hair on the scalp was then removed by massaging the scalp with a depilatory cream (Nair) using a cotton-tipped applicator and leaving the cream on the scalp for 4–5 min. The cream was then removed by a thin plastic sheet (flexible ruler) to leave a hair-free area on the scalp. The remaining tiny hairs were then removed by alcohol swab and the area was then sterilized by applying 10% povidone-iodine (Dynarex, USA) using a sterile cotton-tipped applicator. The medial incision was made with a scalpel blade #10, and 0.2 ml of 0.5% lidocaine was injected intradermally into the scalp. The skin extending from the medial line to the temporalis muscle was completely removed using a pair of microscissors to produce a wide skinless area above the skull. Number 5/45 forceps were used to remove any remaining periosteum. The remaining dried or firmly attached pieces of periosteum were removed with a scalpel blade #10. The skull was cleaned with sterile saline and dried with gently pressurized air. Using the stereotaxic apparatus, a wide area of ~3 × 4 mm$^2$ above the left IC was made. A micro drill bit (size #80, Grainger, USA) was used to drill through the skull starting from the rostrolateral (*Figure 1A*, yellow circle) region to lambda (*Figure 1A*, black circle) following the border in a clockwise direction (*Figure 1A*, dotted black rectangle). To prevent overheating of the superficial brain tissues and to mitigate the occasional spurts of skull bleeding during the drilling, ice-cold sterile saline was used to intermittently irrigate the surface. A stream of pressurized air was also applied during the drilling procedures to prevent overheating and remove the debris produced by the drilling. Caution was taken not to pierce the dura when performing the craniotomy while crossing the sagittal or the lambdoid sutures to avoid damaging the underlying sinuses. After drilling, the skull was irrigated in sterile saline and the bone flap (the undrilled bone over the craniotomy area) was gently examined for complete separation from the rest of the skull (*Figure 1B*). Using a pair of no. 5/45 forceps, the bone flap was gently removed. To control the bleeding if it occurred, a piece of sterile hemostatic gel (Adsorbable Gelatin Sponge USP, Haemosponge, GBI, India), which was pre-soaked in ice-cold saline, was applied to the bleeding spot. Once the bleeding ceased, the brain was kept covered in sterile saline. In some surgeries, the dura was peeled off from the surface of the IC while removing the bone flap. In the surgeries where the dura remained intact, a Bonn microprobe (FST, Germany, Item # 10032-13) was used to pierce the dura in an area that is not above the IC and devoid of cortical vasculature (e.g. a part of exposed cerebellum). After piercing, the dura was carefully and gently lifted, and a pair of no. 5/45 forceps were used to grab the dura to gently tear it to the extent of the transverse sinus to avoid bleeding of the major venous structures.

Using the stereotaxic apparatus, a 30° diamond knife (FST, Germany, Item # 10100-30) was placed in its vertical position. From the medial line, the knife was placed at the lateral horizon of the IC. The knife was then inserted 1.5 mm ventrally. Caution was taken to avoid hitting any major blood vessels to avoid bleeding. From its position, the knife was retracted 0.3–0.5 mm medially to create an initial pocket for the insertion of the microprism (*Figure 1C–E*). The microprism (1.5 mm silver coated, Tower Optical Corporation) was then inserted, so one of its sides was parallel to the lateral surface of the diamond blade, the other side was facing up, and its hypotenuse was facing ventrally (*Figure 1F and G*). The wooden piece of a cotton swab was carefully driven down to push the microprism until its top surface was at the same level as the IC dorsal surface. Keeping the microprism in its place with the wooden end of a swab, the diamond knife was carefully retracted upward leaving the microprism behind (*Figure 1H*). If the retraction of the diamond knife pulled the prism up, the wooden piece would be used to push it down simultaneously with the knife movement. To control the bleeding, a piece of sterile hemostatic gel was used as before. The tip of the diamond knife was used to secure

the microprism in its position for at least 10 min (*Figure 1J*). During that time, the solution of the 1% agarose was prepared for use. A pair of forceps were then used to gently place a sterile 5 mm cover glass #1 (Thomas Scientific, USA) over the skull covering the opened area above the IC. The cover glass was secured by a wooden trimmed piece of sterile cotton swab by gently pressing the cover glass from the top. Since the surface of the IC was located below the surfaces of both the cerebellum and cerebral cortex, the cover glass usually traps about a 0.7–1.0 mm thick sheet of sterile saline. The movement of the liquid medium caused by heart pulsation was found to be a source of motion artifact during imaging, so the gap between the surface of the IC and the cover glass was filled with 1% agarose gel that was made in saline at a temperature range of 33–35 °C (*Figure 1K*). The extra agarose gel pieces were cut off and removed by the scalpel blade and the wet skull was then dried out using pressurized air. A titanium headpost as described before (*Goldey et al., 2014*) was glued carefully on the top of the skull to be at the same level as the cover glass. Following the manufacturer's instructions, the C&B Metabond (Parkell, Japan) was used to secure the headpost in its place. Excluding the steps of the microprism insertion, the same procedures were followed for DC imaging from the surface (*Figure 1L*). For awake preparations, the same procedures were followed under isoflurane anesthesia starting with 4% of isoflurane as an induction dose with 1–2% isoflurane during the surgery. The animal was transferred under the microscope objective and left to recover from isoflurane anesthesia for 1 hr before imaging. To ensure the animal's recovery, the animal was visually inspected for its breathing rate (*Ewald et al., 2011*) and ongoing limb movements.

## Acoustic stimulation

Using a custom-made MATLAB (The MathWorks, Natick, MA, USA) code, either 500 ms pure tone or AM-noise was generated. Thirty-five (5×7) pure tones with a series of sound pressure levels (40, 50, 60, 70, 80 dB SPL) and a series of carrier frequencies (5000–40,000 Hz with a half-octave gap) were presented with a cosine window. Forty-five (5×9) 100% AM white noise bursts with a series of sound pressure levels (40, 50, 60, 70, 80 dB SPL) and modulation frequencies (0, 2, 4, 8, 16, 32, 64, 128, 256 Hz) were generated. The stimuli of either pure tone or AM-noise combinations were played in random sequence to the mice with a 600 ms interstimulus interval (ISI) by a TDT RP2.1 processor (Tucker-Davis Technologies, USA) and delivered by a TDT ES1 speaker (Tucker-Davis Technologies, USA). In other sets of experiments, 500 ms of unmodulated broadband noise was presented five times with ISI of 600 ms at 80 dB SPL to examine the acoustic activity between modules and matrix compared to somatosensory stimulation.

The output of the TDT ES1 speaker was calibrated using a PCB 377A06 microphone, which feeds a SigCal tool to generate a calibration file for all tested frequencies (5–40 kHz). To enable the custom-made MATLAB code to read this calibration file, the values were first processed by MATLAB signal processing toolbox (sptool) to generate a 256-tap FIR filter to apply the calibration using the following parameters: arbitrary magnitudes, least square, order: 256, sampling rate: 97,656.25, frequency vector (5–40 kHz), amplitude vector (40–80 dB SPL), and weight vector ones (1128).

## Somatosensory stimulation

The somatosensory stimulation was done by deflecting the right whiskers of the animals at an average excursion of 1.2 mm with a 50 Hz rate over 500 ms to match the acoustic stimulus. A total of five stimuli were presented with 600 ms ISI. The whisker deflection was performed using a Brüel & Kjær vibrator (V203 10/32 UNF-CE, UK) that was driven by a wave generator (4063, B&K Precision, USA) after amplifying the signals with an SLA1 100 W power amplifier (ART, USA).

## 2P imaging

Immediately after surgery, the anesthetized animal was taken and secured under the microscope objective by clamping the arms of the head post to two perpendicular metal posts mounted on the microscope stage. A custom-built 2P microscope was used. The optical and the controlling components were supplied from Bruker, Olympus, and Thorlabs. The imaging of the DC was made using a ×20 water-immersion objective (LUMPlanFl/IR, ×20, NA: 0.95, WD: 2 mm; Olympus Corporation, Tokyo, Japan), while the lateral surface of the IC was imaged by the long working distance ×16 water-immersion objective (N16XLWD-PF – CFI LWD Plan Fluorite Objective, ×16, NA: 0.8, WD: 3.0 mm; Nikon, Tokyo, Japan) to be able to reach the focal point of the microprism at its hypotenuse. For

imaging both the GFP or jRGECO1a signals, the excitation light was generated by InSight X3 laser (Spectra-Physics Lasers, Mountain View, CA, USA) tuned to a wavelength of 920 or 1040 nm, respectively. A layer of a 1:1 mixture of wavelength multipurpose ultrasound gel (National Therapy, Canada) with double-deionized water was used to immerse the objective. This gel was able to trap the water and reduce its evaporation during imaging. The emitted signals were detected by a photomultiplier tube (Hamamatsu H7422PA-4, Japan) following a t565lp dichroic and a Chroma barrier et525/70m filter for GFP and et595/50m filter for jRGECO1a signals. Images (512×512 pixels) were collected at a frame rate of 29.9 Hz in the resonant galvo mode. Data were collected from the dorsal surface of the IC by scanning the surface of the IC based on the GFP and jRGECO1a signals through the medial and lateral horizons of the IC. Generally, the scanning was started by moving the ×20 objective to the most ventromedial position, where there is no ability to see any cells (*Figure 2B*, medial horizon). Then, the objective was driven to slowly move dorsolaterally to get cells in focus (*Figure 1M*), where the average depth of the first field of view was 169 μm. The lateral movement of the objective was intended to be aligned and parallel to the curvature of the DC surface indicated by getting the superficial cells in focus to avoid any possibility of imaging the upper layers of the central nucleus of the IC, so the average depth of the most superficial point of the imaging was 92 μm. The scanning was ended by moving the objective to the most ventrolateral position of the IC, where there is no ability to see GFP laterally (*Figure 2B*, lateral horizon), where the last field of view was collected at an average depth of 225 μm. Generally, each field of view was selected based on the expression of GFP signals as an indicator for the cells in focus and being acoustically active by using a search stimulus that was 500 ms broadband noise with zero modulation at 80 dB SPL. The z-axis of the objective was determined relative to the pia surface. The relative position of each region was tracked through the micromanipulator (MP-285, Sutter Instruments) that controlled the microscope objective. The positions of imaged areas were further aligned across animals to a common coordinate using the midline, lateral extremes, and the vasculature landmarks taken via imaging of the GFP signals of the cranial window with low magnification (×4 Olympus Objectives, 4×4 binning, 50 ms frame time) using a Lumen 200 bulb (Prior Scientific Inc, USA) and CoolSNAP MYO camera (Photometrics, USA, excitation: 488 nm and emission: 515–550 nm). Imaging of the lateral surface of the IC via microprism was done similarly. In both cases, the frame timing of the scanner and the sound stimuli were both digitized and time-locked using a Digidata 1440A (Molecular Devices, Sunnyvale, CA, USA) with Clampex v. 10.3 (Molecular Devices, Sunnyvale, CA, USA).

## Laser lesion

The laser lesion was produced using a 720 or 820 nm laser beam at 74–80 mW power at the level of the objective for 100–250 s depending on the size of the lesioned area, the speed of laser scanning (frame rate), and the depth of the lesion (z-axis length). In brief, the area of interest and the dimension of the laser lesion were initially determined using 920 nm as indicated by the GFP signals through scanning one of the GABAergic modules. The depth of the GABAergic modules was then determined by counting the optical slices starting from their superficial surface. The tunable laser was then switched from 920 nm to the shorter wavelength (720 or 820 nm) to scan the same area and depth at a rate 1–2 frames/s.

## Data processing
### Data collection

The data were collected as separate movies (512×512 pixels) for each pure tone or AM-noise runs in a resonant galvo mode. Depending on the amplitude and frequency combinations for each type of acoustic stimulus, 40 or 52 s periods were assigned as a movie's length for pure tone (35 stimulus combinations) or AM-noise (45 stimulus combinations), respectively. Using ImageJ software, the z-projection was used to compute one single image representing either the sum, the standard deviation, or the median of all the image sequences in the movie. Based on these single images, the region of interest (ROI) was manually drawn around each detectable cell body. *Figure 3A and B* represents the sum of the image sequences of the movie collected from the dorsal surface, or the lateral surface of the IC via microprism, respectively. For the subsequent processing steps, Python open-source libraries were used (listed below).

## Motion correction and filtering

The imread function from the OpenCV library was used in grayscale mode to import images into a numpy array from a single folder containing TIFF images. The array was then exported as a single TIFF stack into a temporary folder using the mimwrite function from the imageio library, and the process was repeated for each folder. The NoRMCorre algorithm (*Pnevmatikakis and Giovannucci, 2017*) embedded in the CaImAn library (*Giovannucci et al., 2019*) was used to apply motion correction to each of the TIFF files. The data were then imported into a numpy array, rounded, and converted to 16-bit integers. The images were filtered using a 2D Gaussian filter with a sigma value of 1 (surface View)/2 (prism View) in each direction, then a 1D Gaussian temporal filter with a sigma value of 2 was applied using the ndimage.gaussian_filter and ndimage.gaussian_filter1d function from the scipy library, respectively.

## Data extraction

The ROI sets, which were manually created using ImageJ, were imported using the read_roi_zip function from the read_roi library. The sets were then used to create two masks: one mask was used as a replica of the ROIs and the second mask was made around the original ROI (roughly four times larger in area). The smaller mask was applied to find the average pixel value within each ROI, while the larger mask was applied to find the average pixel value of the neuropil. The neuropil correction was applied using the following equation (*Akerboom et al., 2012*; *Kerlin et al., 2010*):

$$\text{Corrected value} = \text{Date value} - \left( 0.4 \times \text{Neuropil value} \right)$$

To identify the calcium signals, Δf/f was calculated by using the following equation:

$$\frac{\Delta\text{f}}{\text{f}} = \frac{\left( \text{Date value} - \text{Background value} \right)}{\text{Background value}}$$

where the background value is the slope estimating the background levels with fluorescence bleaching factored in. The data were then reorganized so that all segments with the same stimulus frequency and stimulus amplitude were grouped. The AUC of the calcium signals of the excitatory responses was only then used as a metric to determine the magnitude of the response and for subsequent analysis (*Berens et al., 2018*; *Overk et al., 2015*).

## Cell flagging

The correlation coefficient between each of the trials was calculated using the stats. Pearson function from the scipy library. The average correlation coefficient was calculated for each stimulus frequency and stimulus amplitude. Similar to previous work (*Wong and Borst, 2019*), if the average correlation coefficient was above the threshold of 0.6, the cell was flagged as being responsive to that stimulus combination of frequency and amplitude, and the best tone and modulation frequencies were calculated for every cell (*Barnstedt et al., 2015*). Knowing BTF and the best modulation frequency (BMF) for each neuron enabled us to calculate the SMI and TMI at each sound level following these equations:

$$\text{SMI} = \frac{\text{Response to BTF} - \text{Response to unmodulated noise}}{\text{Response to BTF} + \text{Response to unmodulated noise}}$$

The SMI has values ranging from −1 to 1. While neurons with values closer to 1 are more responsive to pure tone, neurons with values closer to –1 are more responsive to noise with no modulation.

$$\text{TMI} = \frac{\text{Response to BMF} - \text{Response to unmodulated noise}}{\text{Response of BMF} + \text{Response to unmodulated noise}}$$

Similarly, the TMI has values ranging from −1 to 1. While neurons with values closer to 1 are more responsive to AM-noise, neurons with values closer to –1 are more responsive to noise with no modulation.

For the comparison between somatosensory and auditory responses, the threshold of 0.4 was used as a cutoff threshold to determine the responsive cells to each stimulation. The average response of

each somatosensory, acoustic (500 ms of unmodulated noise), or simultaneous somatosensory and acoustic (bimodal) stimulation was then calculated for each responsive cell. The correlation method was validated as a tool to determine the responsive cells to somatosensory stimulation by generating an ROC curve comparing the automated method to a blinded human interpretation. The AUC of the ROC curve was 0.88. This high AUC value indicates that the correlation method can rank the random responsive cells than the random nonresponsive cells. At the correlation coefficient (0.4), which was the cutoff value to determine the responsive cells for somatosensory stimulation, the specificity was 87% and the sensitivity 72%, the positive predictive value was 73%, and the negative predictive value was 86%. Additionally, all the false-responsive cells were excluded from the analysis. To examine the effect of somatosensory stimulation on auditory responses, the RI was calculated for the auditory-responsive cells using the following equation:

$$RI = \frac{\text{Auditory response to bimodal stimulation}}{\text{Auditory response to auditory stimulation only}}$$

Neurons with RI values>1 have enhanced or higher responses induced by simultaneous auditory and somatosensory stimulation, while neurons with RI values<1 have downregulated or suppressed auditory responses induced by auditory and simultaneous somatosensory stimulation.

## Map generation

The average radius for all ROIs was calculated to ensure that all cells on the tonotopic map had uniform radii. A color key was also generated for every corresponding map. A blank canvas was generated using the Image module from the pillow library, and a circle was created for each cell and filled with a corresponding color to its value.

## RFS analysis

The binary RFS was calculated for every cell by determining whether the cell was responsive to a particular combination of frequency and amplitude (*Bowen et al., 2020*). A value of 1 was given to the responsive cell, if it responds to a single frequency/amplitude combination. In contrast, a value of 1 was added to the tally for each responsive combination if the cell was broadly responsive. Accordingly, the higher and lower values of RFS indicate a broad and narrow RF, respectively. To examine the differences between the RFS of the cells at different areas under different preparations, the cumulative distribution function was used.

## Assessment of tonotopic organization

The tonotopy was assessed by examining the best fit of linear or nonlinear quadratic polynomial regressions between the BTFs of cells and their locations along different anatomical axes. For the $LC_{(microprism)}$, the regression fit was examined along the main four axes such as dorsal to ventral, rostral to caudal, dorsocaudal to ventrorostral, and dorsorostral to ventrocoudal. Based on the previous reports showing the tonotopic organization over the DC (*Barnstedt et al., 2015*; *Wong and Borst, 2019*), the tonotopy was examined along the rostromedial to caudolateral axis. The $x_2$ and $y_2$ coordinates of each cell were determined. At the starting point of each axis, the $x_1$ and $y_1$ coordinates were assigned values as (0, 0). Then, the distance (d) between each cell and the starting point at each axis was calculated based on the following equation:

$$d = \sqrt{(x2 - x1)^2 + (y2 - y1)^2}$$

For each 50 µm (bin size), the geometric average of the BTFs of all cells as well as their distances from the starting point at each axis were computed. The linear or quadratic polynomial regressions were then examined between the geometric averages of BTFs and the geometric averages of distances of cells for each animal.

## Detecting the spontaneous activity from the nonresponsive cells

Movies have been recorded from the nonresponsive cells of the $LC_{(microprism)}$ when no stimulation was presented. The time trace of jRGECO1a signals was extracted from each cell. A polynomial fitting of the time trace was used to correct for photobleaching and to adjust for the baseline for the whole

trace. The sliding average of the baseline was then calculated for 1 s (~29 frames), and the z-score was calculated for the subsequent frames. The spontaneous activity was detected when the z-score was >3.0.

## Statistics

All statistical analyses and graphs were made using Origin Pro software. The normality of the distributions of the data was examined using a Kolmogorov-Smirnov test. The Mann-Whitney test as well as two-sample or paired t-test were used to examine the difference in data distribution or the difference between samples of non-normal and normal distribution, respectively. Either one-way or two-way ANOVA as well as Fisher as a post hoc test were used for the mean comparison between different groups of normal distribution. The chi-square ($\chi^2$) test or Kruskal-Wallis ANOVA tests were used to compare the samples based on cell numbers of their cellular population. The cumulative frequency function was computed to calculate the fraction of cells tuned at each frequency based on their best frequency. Differences were deemed significant when p-value<0.05. The upper and lower borders of each box plot are at the 25th and 75th percentile, with whiskers at 95% spread.

## Histology

After the 2P imaging, the mouse was transcardially perfused with 4% paraformaldehyde (PFA) solution in PBS solution. The isolated brain was then placed overnight in PFA for a complete fixation. The brain was then moved from the PFA to a graded sucrose solution (10%, 20%, and 30%) until it sunk. 50 µm coronal sections were taken at the level of the IC using a Leica cryostat. The sections were then mounted on a slide, coverslipped, and imaged using a confocal microscope for imaging the GFP signals as an indicator for the GABAergic cells in the GAD67-GFP knock-in mouse (excitation: 488 nm and emissions: 515–550 nm).

For the examination of jrgeco1a expression in the IC, the mouse was transcardially perfused with saline with a high concentration of potassium and a high ratio of calcium to magnesium ([in mM] 3.5 KCl, 25 glucose, 125 NaCl, 1.2 $KH_2PO_4$, 2 $CaCl_2$, 0.5 $MgSO_4$) under ketamine anesthesia. The brain was then fixed by a subsequent perfusion of 4% PFA. Selected 40 µm coronal sections at the level of the IC were immunostained for NeuN using (NeuN [D4G40] XP Rabbit mAB, Cell Signaling, Cat# 62994, 1:100) primary antibody and (goat anti-rabbit IgG conjugated to Alexa Fluor 405, 1:200, Thermo Fisher, USA) secondary antibody following the immunostaining procedures that were previously described (*Issa et al., 2023*; *Vaithiyalingam Chandra Sekaran et al., 2021*). Confocal images were obtained using a Confocal Zeiss LSM 710 Microscope using ×10 and ×40 objectives and tile scan functions. For NeuN, Alexa Fluor 405 signals were obtained by a 405 nm laser and emission at 415–466 nm. The GFP signals were obtained by a 488 nm laser and emission at 501–551 nm. The jRGECO1a signals were obtained by 561 nm laser and emission at 600–700 nm. All cells with somas expressing a red fluorescence of jrgeco1a were quantified in 2P-imaged areas of the DC and LC. The density of the cells expressing jrgeco1a was then calculated based on the cells expressing NeuN.

## Artwork

The cartoon of the mouse brain in *Figure 1C, F, and I* was obtained from Brain Explorer 3D viewer Allen Mouse Brain (https://connectivity.brain-map.org/3d-viewer?v=1) and modified to be suitable for illustration. All figures were designed and made using Adobe Illustrator (Adobe, San Jose, CA, USA). To keep working within the Adobe environment to avoid losing the resolution of the figures, Adobe Photoshop (Adobe, San Jose, CA, USA) was used to crop the borders of some images to save space. The color pallets were chosen to avoid the combination of the green and red colors for all maps, so they will be suitable for color-blind readers. To induce the same effect in the microscope images combining the red (jRGECO1a) and green (GFP) fluorescence, we kept the green signals of the GFP unchanged, but we substituted the red fluorescence of jRGECO1a with magenta as a pseudocolor.

## Acknowledgements

We thank Dr. Kush Paul (Bruker) for initial training on the craniotomy surgery, Dr. Jason Maclean (The Department of Neurobiology, University of Chicago) for assistance with 2P imaging, Dr. Michael Jacobs Goard (Neuroscience Research Institute, The University of California- Santa Barbara) for his guidance in the use of microprisms, Dr. Taher Saif (Department of Mechanical Science and Engineering, The

University of Illinois Urbana-Champaign) for measuring the deflecting distance of the somatosensory stimulating probe using a high-speed camera, Dr. Pengfei Song (Department of Electric and Computer Engineering, The University of Illinois Urbana-Champaign) for his assistance with the somatosensory stimulating probe, and Dr. Nathiya Vaithiyalingam Chandra Sekaran (Department of Molecular and Integrative Physiology, The University of Illinois Urbana-Champaign) for her assistance in immuno-histochemistry and confocal imaging. We also thank the NIDCD for funding this work under grants # R01DC013073, R01DC016599, and R21DC021605.

## Additional information

### Funding

| Funder | Grant reference number | Author |
|---|---|---|
| National Institute on Deafness and Other Communication Disorders | R01DC016599 | Daniel A Llano |
| National Institute on Deafness and Other Communication Disorders | R01DC013073 | Daniel A Llano |
| National Institute on Deafness and Other Communication Disorders | R21DC021605 | Daniel A Llano |

The funders had no role in study design, data collection and interpretation, or the decision to submit the work for publication.

### Author contributions

Baher A Ibrahim, Conceptualization, Data curation, Formal analysis, Investigation, Methodology, Writing – original draft, Writing – review and editing; Yoshitaka Shinagawa, Austin Douglas, Formal analysis, Visualization, Methodology; Gang Xiao, Formal analysis, Investigation, Methodology; Alexander R Asilador, Investigation, Methodology; Daniel A Llano, Conceptualization, Resources, Data curation, Funding acquisition, Investigation, Methodology, Writing – original draft, Project administration, Writing – review and editing

### Author ORCIDs

Baher A Ibrahim ⓘ https://orcid.org/0000-0002-0062-7589
Daniel A Llano ⓘ https://orcid.org/0000-0003-0933-1837

### Ethics

This study was performed in strict accordance with the recommendations in the Guide for the Care and Use of Laboratory Animals of the National Institutes of Health. All of the animals were handled according to approved institutional animal care and use committee (IACUC) protocols (#21214) of the University of Illinois at Urbana-Champaign (AAALAC #00766). All surgery was performed under general anesthesia, and every effort was made to minimize suffering.

Reviewer #1 (Public Review): https://doi.org/10.7554/eLife.93063.4.sa1
Reviewer #2 (Public Review): https://doi.org/10.7554/eLife.93063.4.sa2
Reviewer #3 (Public Review): https://doi.org/10.7554/eLife.93063.4.sa3
Author response https://doi.org/10.7554/eLife.93063.4.sa4

## Additional files

### Supplementary files

MDAR checklist

## Data availability

The numerical data points of some graphs and the code used for generating the cell responses and maps are publicly available on Dryad (https://doi.org/10.5061/dryad.v41ns1s70).

The following dataset was generated:

| Author(s) | Year | Dataset title | Dataset URL | Database and Identifier |
|---|---|---|---|---|
| Shinagawa Y, Douglas A, Xiao G, Asilador A, Llano DA | 2025 | Data from: Microprism-based two-photon imaging of the mouse inferior colliculus reveals novel organizational principles of the auditory midbrain | https://doi.org/10.5061/dryad.v41ns1s70 | Dryad Digital Repository, 10.5061/dryad.v41ns1s70 |

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
