## [Editor Report · eLife assessment]

This study provides **valuable** new insights into how multisensory information is processed in the lateral cortex of the inferior colliculus, a poorly understood part of the auditory midbrain. By developing new imaging techniques that provide the first optical access to the lateral cortex in a living animal, the authors provide **convincing** in vivo evidence that this region contains separate subregions that can be distinguished by their sensory inputs and neurochemical profiles, as suggested by previous anatomical and in vitro studies. This work provides a foundation for future research exploring how this part of the auditory midbrain contributes to multisensory-based behavior.

---

## [Referee Report · Reviewer #1 (Public Review)]

In this paper the authors provide a characterisation of auditory responses (tones, noise, and amplitude modulated sounds) and bimodal (somatosensory-auditory) responses and interactions in the higher order lateral cortex (LC) of the inferior colliculus (IC) and compare these characteristic with the higher order dorsal cortex (DC) of the IC - in awake and anaesthetised mice. Dan Llano's group have previously identified gaba'ergic patches (modules) in the LC distinctly receiving inputs from somatosensory structures, surrounded by matrix regions receiving inputs from auditory cortex. They here use 2P calcium imaging combined with an implanted prism to - for the first time - get functional optical access to these subregions (modules and matrix) in the lateral cortex of IC in vivo, in order to also characterise the functional difference in these subparts of LC. They find that both DC and LC of both awake and anaesthetised appears to be more responsive to more complex sounds (amplitude modulated noise) compared to pure tones and that under anesthesia the matrix of LC is more modulated by specific frequency and temporal content compared to the gaba'ergic modules in LC. However, while both LC and DC appears to have low frequency preferences, this preference for low frequencies is more pronounced in DC. Furthermore, in both awake and anesthetized mice somatosensory inputs are capable of driving responses on its own in the modules of LC, but very little in the matrix. The authors now compare bimodal interactions under anaesthesia and awake states and find that effects are different in some cases under awake and anesthesia - particularly related to bimodal suppression and enhancement in the modules.

The paper provides new information about how subregions with different inputs and neurochemical profiles in the higher order auditory midbrain process auditory and multisensory information, and is useful for the auditory and multisensory circuits neuroscience community.

---

## [Referee Report · Reviewer #2 (Public Review)]

Summary:

The study describes differences in responses to sounds and whisker deflections as well as combinations of these stimuli in different neurochemically defined subsections of the lateral and dorsal cortex of the inferior colliculus in anesthetised and awake mice.

Strengths:

A major achievement of the work lies in obtaining the data in the first place as this required establishing and refining a challenging surgical procedure to insert a prism that enabled the authors to visualise the lateral surface of the inferior colliculus. Using this approach, the authors were then able to provide the first functional comparison of neural responses inside and outside of the GABA-rich modules of the lateral cortex. The strongest and most interesting aspects of the results, in my opinion, concern the interactions of auditory and somatosensory stimulation. For instance, the authors find that (a) somatosensory-responses are strongest inside the modules and (b) somatosensory-auditory suppression is stronger in the matrix than in the modules. This suggests that, while somatosensory inputs preferentially target the GABA-rich modules, they do not exclusively target GABAergic neurons within the modules (given that the authors record exclusively from excitatory neurons we wouldn't expect to see somatosensory responses if they targeted exclusively GABAergic neurons) and that the GABAergic neurons of the modules (consistent with previous work) preferentially impact neurons outside the modules, i.e. via long-range connections.

---

## [Referee Report · Reviewer #3 (Public Review)]

The lateral cortex of the inferior colliculus (LC) is a region of the auditory midbrain noted for receiving both auditory and somatosensory input. Anatomical studies have established that somatosensory input primarily impinges on "modular" regions of the LC, which are characterized by high densities of GABAergic neurons, while auditory input is more prominent in the "matrix" regions that surround the modules. However, how auditory and somatosensory stimuli shape activity, both individually and when combined, in the modular and matrix regions of the LC has remained unknown.

The major obstacle to progress has been the location of the LC on the lateral edge of the inferior colliculus where it cannot be accessed in vivo using conventional imaging approaches. The authors overcame this obstacle by developing methods to implant a microprism adjacent to the LC. By redirecting light from the lateral surface of the LC to the dorsal surface of the microprism, the microprism enabled two-photon imaging of the LC via a dorsal approach in anesthetized and awake mice. Then, by crossing GAD-67-GFP mice with Thy1-jRGECO1a mice, the authors showed that they could identify LC modules in vivo using GFP fluorescence while assessing neural responses to auditory, somatosensory, and multimodal stimuli using Ca2+ imaging. Critically, the authors also validated the accuracy of the microprism technique by directly comparing results obtained with a microprism to data collected using conventional imaging of the dorsal-most LC modules, which are directly visible on the dorsal IC surface, finding good correlations between the approaches.

Through this innovative combination of techniques, the authors found that matrix neurons were more sensitive to auditory stimuli than modular neurons, modular neurons were more sensitive to somatosensory stimuli than matrix neurons, and bimodal, auditory-somatosensory stimuli were more likely to suppress activity in matrix neurons and enhance activity in modular neurons. Interestingly, despite their higher sensitivity to somatosensory stimuli than matrix neurons, modular neurons in the anesthetized prep were overall more responsive to auditory stimuli than somatosensory stimuli (albeit with a tendency to have offset responses to sounds). This suggests that modular neurons should not be thought of as primarily representing somatosensory input, but rather as being more prone to having their auditory responses modified by somatosensory input. However, this trend was different in the awake prep, where modular neurons became more responsive to somatosensory stimuli. Thus, to this reviewer, one of the most intriguing results of the present study is the extent to which neural responses in the LC changed in the awake preparation. While this is not entirely unexpected, the magnitude and stimulus specificity of the changes caused by anesthesia highlight the extent to which higher-level sensory processing is affected by anesthesia and strongly suggests that future studies of LC function should be conducted in awake animals.

Together, the results of this study expand our understanding of the functional roles of matrix and module neurons by showing that responses in LC subregions are more complicated than might have been expected based on anatomy alone. The development of the microprism technique for imaging the LC will be a boon to the field, finally enabling much-needed studies of LC function in vivo. The experiments were well-designed and well-controlled, the limitations of two-photon imaging for tracking neural activity are acknowledged, and appropriate statistical tests were used.

---

## [Author Response]

The following is the authors’ response to the previous reviews.

**Public Reviews:**

**Reviewer #1 (Public Review):**
In this paper the authors provide a characterisation of auditory responses (tones, noise, and amplitude modulated sounds) and bimodal (somatosensory-auditory) responses and interactions in the higher order lateral cortex (LC) of the inferior colliculus (IC) and compare these characteristic with the higher order dorsal cortex (DC) of the IC - in awake and anaesthetised mice. Dan Llano's group have previously identified gaba'ergic patches (modules) in the LC distinctly receiving inputs from somatosensory structures, surrounded by matrix regions receiving inputs from auditory cortex. They here use 2P calcium imaging combined with an implanted prism to - for the first time - get functional optical access to these subregions (modules and matrix) in the lateral cortex of IC in vivo, in order to also characterise the functional difference in these subparts of LC. They find that both DC and LC of both awake and anaesthetised appears to be more responsive to more complex sounds (amplitude modulated noise) compared to pure tones and that under anesthesia the matrix of LC is more modulated by specific frequency and temporal content compared to the gaba'ergic modules in LC. However, while both LC and DC appears to have low frequency preferences, this preference for low frequencies is more pronounced in DC. Furthermore, in both awake and anesthetized mice somatosensory inputs are capable of driving responses on its own in the modules of LC, but very little in the matrix. The authors now compare bimodal interactions under anaesthesia and awake states and find that effects are different in some cases under awake and anesthesia - particularly related to bimodal suppression and enhancement in the modules.The paper provides new information about how subregions with different inputs and neurochemical profiles in the higher order auditory midbrain process auditory and multisensory information, and is useful for the auditory and multisensory circuits neuroscience community.The manuscript is improved by the response to reviewers. The authors have addressed my comments by adding new figures and panels, streamlining the analysis between awake and anaesthetised data (which has led to a more nuanced, and better supported conclusion), and adding more examples to better understand the underlying data. In streamlining the analyses between anaesthetised and awake data I would probably have opted for bringing these results into merged figures to avoid repetitiveness and aid comparison, but I acknowledge that that may be a matter of style. The added discussions of differences between awake and anaesthesia in the findings and the discussion of possible reasons why these differences are present help broaden the understanding of what the data looks like and how anaesthesia can affect these circuits.As mentioned in my previous review, the strength of this study is in its demonstration of using prism 2p imaging to image the lateral shell of IC to gain access to its neurochemically defined subdivisions, and they use this method to provide a basic description of the auditory and multisensory properties of lateral cortex IC subdivisions (and compare it to dorsal cortex of IC). The added analysis, information and figures provide a more convincing foundation for the descriptions and conclusions stated in the paper. The description of the basic functionality of the lateral cortex of the IC are useful for researchers interested in basic multisensory interactions and auditory processing and circuits. The paper provides a technical foundation for future studies (as the authors also mention), exploring how these neurochemically defined subdivisions receiving distinct descending projections from cortex contribute to auditory and multisensory based behaviour.Minor comment:- The authors have now added statistics and figures to support their claims about tonotopy in DC and LC. I asked for and I think allows readers to better understand the tonotopical organisation in these areas. One of the conclusions by the authors is that the quadratic fit is a better fit that a linear fit in DCIC. Given the new plots shown and previous studies this is likely true, though it is worth highlighting that adding parameters to a fitting procedure (as in the case when moving from linear to quadratic fit) will likely lead to a better fit due to the increased flexibility of the fitting procedure.

Thank you for the suggestion. We have highlighted that the quadratic function allowed the regression model to include the cells tuned to higher frequencies at the rostromedial part of the DC and result in a better fit, which is consistent with the tonotopic organization that was previously described as shown in text at (lines 208-211).

**Reviewer #2 (Public Review):**
Summary:The study describes differences in responses to sounds and whisker deflections as well as combinations of these stimuli in different neurochemically defined subsections of the lateral and dorsal cortex of the inferior colliculus in anesthetised and awake mice.Strengths:A major achievement of the work lies in obtaining the data in the first place as this required establishing and refining a challenging surgical procedure to insert a prism that enabled the authors to visualise the lateral surface of the inferior colliculus. Using this approach, the authors were then able to provide the first functional comparison of neural responses inside and outside of the GABA-rich modules of the lateral cortex. The strongest and most interesting aspects of the results, in my opinion, concern the interactions of auditory and somatosensory stimulation. For instance, the authors find that (a) somatosensory-responses are strongest inside the modules and (b) somatosensory-auditory suppression is stronger in the matrix than in the modules. This suggests that, while somatosensory inputs preferentially target the GABA-rich modules, they do not exclusively target GABAergic neurons within the modules (given that the authors record exclusively from excitatory neurons we wouldn't expect to see somatosensory responses if they targeted exclusively GABAergic neurons) and that the GABAergic neurons of the modules (consistent with previous work) preferentially impact neurons outside the modules, i.e. via long-range connections.Weaknesses:While the findings are of interest to the subfield they have only rather limited implications beyond it and the writing is not quite as precise as it could be.
**Reviewer #3 (Public Review):**
The lateral cortex of the inferior colliculus (LC) is a region of the auditory midbrain noted for receiving both auditory and somatosensory input. Anatomical studies have established that somatosensory input primarily impinges on "modular" regions of the LC, which are characterized by high densities of GABAergic neurons, while auditory input is more prominent in the "matrix" regions that surround the modules. However, how auditory and somatosensory stimuli shape activity, both individually and when combined, in the modular and matrix regions of the LC has remained unknown.The major obstacle to progress has been the location of the LC on the lateral edge of the inferior colliculus where it cannot be accessed in vivo using conventional imaging approaches. The authors overcame this obstacle by developing methods to implant a microprism adjacent to the LC. By redirecting light from the lateral surface of the LC to the dorsal surface of the microprism, the microprism enabled two-photon imaging of the LC via a dorsal approach in anesthetized and awake mice. Then, by crossing GAD-67-GFP mice with Thy1-jRGECO1a mice, the authors showed that they could identify LC modules in vivo using GFP fluorescence while assessing neural responses to auditory, somatosensory, and multimodal stimuli using Ca2+ imaging. Critically, the authors also validated the accuracy of the microprism technique by directly comparing results obtained with a microprism to data collected using conventional imaging of the dorsal-most LC modules, which are directly visible on the dorsal IC surface, finding good correlations between the approaches.Through this innovative combination of techniques, the authors found that matrix neurons were more sensitive to auditory stimuli than modular neurons, modular neurons were more sensitive to somatosensory stimuli than matrix neurons, and bimodal, auditory-somatosensory stimuli were more likely to suppress activity in matrix neurons and enhance activity in modular neurons. Interestingly, despite their higher sensitivity to somatosensory stimuli than matrix neurons, modular neurons in the anesthetized prep were overall more responsive to auditory stimuli than somatosensory stimuli (albeit with a tendency to have offset responses to sounds). This suggests that modular neurons should not be thought of as primarily representing somatosensory input, but rather as being more prone to having their auditory responses modified by somatosensory input. However, this trend was different in the awake prep, where modular neurons became more responsive to somatosensory stimuli. Thus, to this reviewer, one of the most intriguing results of the present study is the extent to which neural responses in the LC changed in the awake preparation. While this is not entirely unexpected, the magnitude and stimulus specificity of the changes caused by anesthesia highlight the extent to which higher-level sensory processing is affected by anesthesia and strongly suggests that future studies of LC function should be conducted in awake animals.Together, the results of this study expand our understanding of the functional roles of matrix and module neurons by showing that responses in LC subregions are more complicated than might have been expected based on anatomy alone. The development of the microprism technique for imaging the LC will be a boon to the field, finally enabling much-needed studies of LC function in vivo. The experiments were well-designed and well-controlled, the limitations of two-photon imaging for tracking neural activity are acknowledged, and appropriate statistical tests were used.
**Recommendations for the authors:**

**Reviewer #1 (Recommendations For The Authors):**
- Increase font size of scale bars on figure 6.

Thank you for the suggestion. We have increased the font size of the scale bar.

**Reviewer #2 (Recommendations For The Authors):**
Line 505: typo: 'didtinction'

Thank you for the suggestion and we do apologize for the typo. We have fixed the word as shown in the text (line 506).

No further comments.
**Reviewer #3 (Recommendations For The Authors):**
Line 543: Change "contripute" to "contribute"

Thank you for the suggestion and we do apologize for the typo. We have fixed the word as shown in the text (line 544).